# MLRC-BENCH: Can Language Agents Solve Machine Learning Research Challenges?

**Yunxiang Zhang**[α*]    **Muhammad Khalifa**[α]    **Shitanshu Bhushan**[α]    **Grant D Murphy**[α]
**Lajanugen Logeswaran**[β]    **Jaekyeom Kim**[β]    **Moontae Lee**[βγ]    **Honglak Lee**[αβ]    **Lu Wang**[α]
University of Michigan[α]    LG AI Research[β]    University of Illinois at Chicago[γ]

## Abstract

We introduce **MLRC-BENCH**, a benchmark designed to quantify how effectively language agents can tackle challenging **M**achine **L**earning (ML) **R**esearch **C**ompetitions, with a focus on open research problems that demand novel methodologies. Unlike prior work, e.g., AI Scientist [40], which evaluates the end-to-end agentic pipeline by using LLM-as-a-judge, MLRC-BENCH measures the key steps of *proposing* and *implementing* novel research methods and evaluates them with rigorous protocol and objective metrics. Our curated suite of 7 competition tasks reveals significant challenges for LLM agents. Even the best-performing tested agent (gemini-exp-1206 under MLAB [22]) closes only 9.3% of the gap between baseline and top human participant scores. Furthermore, our analysis reveals a misalignment between the *LLM-judged* innovation and their *actual* performance on cutting-edge ML research problems. MLRC-BENCH is a dynamic benchmark, which is designed to continually grow with new ML competitions to encourage rigorous and objective evaluations of AI's research capabilities. Our leaderboard and code are publicly available at `https://huggingface.co/spaces/launch/MLRC_Bench`.

## 1 Introduction

Evaluating large language model (LLM) research agents [2, 35, 40] has so far been restricted to two directions. One involves tasking the agent with end-to-end scientific discovery: proposing a research idea, writing implementation code, running experiments, and eventually producing a full paper as done by AI Scientist [40]. One issue with such evaluation is the lack of reliable baseline method that enables *objective* evaluation of the proposed approach. The second direction, on the other hand, evaluates the agent's ability to produce code that solves a Kaggle-style machine learning (ML) engineering competition, skipping idea proposal and paper writing altogether [22, 7]. While evaluation in this case is straightforward, this setup rarely demands genuine research *novelty* beyond existing methods. Consequently, neither of these setups paints a full picture of whether LLM research agents can design research ideas that are both novel and effective, which we aim to address here.

Competitions at ML conferences and workshops provide a valuable testbed for evaluating research agents by assessing both novelty and effectiveness against established baselines. Unlike Kaggle-style contests, these challenges address unresolved and important problems recognized by the ML community. In addition, public leaderboards facilitate objective comparisons to human experts. If an algorithm truly outperforms known baselines, improvements in benchmark scores provide a reliable signal as to the effectiveness of the proposed method.

Therefore, this paper introduces **MLRC-BENCH** as a benchmark to evaluate the ability of LLM-based research agents to propose and implement novel methods. Drawing on tasks from recent ML conference competitions, MLRC-BENCH enables the evaluation of both **novelty** and **effectiveness**

---

*  Correspondence to `yunxiang@umich.edu`

39th Conference on Neural Information Processing Systems (NeurIPS 2025) Track on Datasets and Benchmarks.

Table 1: 7 MLRC-BENCH tasks representing cutting-edge machine learning research. For each competition, we show the venue where the competition is held, research area, data modality, performance metric, along with the constraints presented to the agents, including maximum allowed runtime and GPU memory based on our hardware configurations. Detailed task descriptions are given in Appendix A.

| Competition | Venue | Research Area | Modality | Metric | Test Runtime | GPU Memory |
|---|---|---|---|---|---|---|
| LLM Merging [52] | NeurIPS 2024 | Efficient LLM | Text | Accuracy, ROUGE | 1 hour | 48 GB |
| Backdoor Trigger Recovery [60] | NeurIPS 2024 | LLM Safety | Text | REASR, Recall | 0.5 hour | 48 GB |
| Temporal Action Localisation [20] | ECCV 2024 Workshop | Multimodal Perception | Video, Audio | mAP | 0.5 hour | 16 GB |
| Rainfall Prediction [19] | NeurIPS 2023 | AI for Science | Satellite Data | Critical Success Index | 0.5 hour | 48 GB |
| Machine Unlearning [54] | NeurIPS 2023 | Data Privacy | Image | Forgetting Quality, Accuracy | 0.5 hour | 16 GB |
| Next Product Recommendation [28] | KDD Cup 2023 | Recommendation System | Text | Mean Reciprocal Rank | 0.5 hour | 16 GB |
| Cross-Domain Meta Learning [6] | NeurIPS 2022 | Few-Shot Learning | Image | Accuracy | 3.5 hours | 16 GB |

of research agents' ideas compared to a reliable baseline method and the top human solution. In particular, it emphasizes objective metrics on tasks such as LLM merging [52] and machine unlearning [54], closely mirroring ongoing research challenges. Moreover, the challenges in MLRC-BENCH can dynamically grow by incorporating new competitions from future ML conferences.

We curate MLRC-BENCH starting with 7 competition tasks in Table 1. We pick tasks that involve novel and high-impact problems, spanning areas including LLM safety, multimodal perception, and few-shot learning. Our experimental findings reveal that even the best-performing tested LLM agents, such as gemini-exp-1206 [46] under the MLAB [22] scaffolding, closes only 9.3% of the gap between baseline and top human participant score. Additionally, our analysis highlights a poor correlation between the novelty judged by LLM and practical effectiveness of agents' solutions, questioning the reliability of LLM-as-a-judge for research idea evaluation. These results underscore the limitations of current AI research agents in generating and implementing innovative ML solutions, providing a crucial benchmark for future advancements.

Our contributions can be summarized as below:

- We introduce MLRC-BENCH, a dynamic benchmark suite curated from ML conference competitions, featuring open research problems that are both impactful and objectively measurable, and that demand the development of novel methodologies;

- We conduct large-scale, objective evaluations for a wide array of frontier LLMs with representative agent scaffoldings, highlighting their inability to propose and implement innovative solutions with notable performance gains;

- We pinpoint the flaws in subjective evaluations of LLM-based research agents, by showing that the LLM-judged idea novelty is misaligned with empirical effectiveness.

## 2 Related Work

Scientific discovery in machine learning typically includes four main stages: **Problem Identification**, where gaps in existing methods are recognized; **Method Proposal**, which introduces a new approach to address the issue; **Experiment Design**, involving the selection of datasets, baselines, and metrics for evaluation; and **Code Implementation**, where the method is realized through executable code. While prior work [40, 49] covers Problem Identification and Experiment Design, evaluation could be subjective based on idea proposal or final paper. Instead, MLRC-BENCH focuses on the critical stages of proposing and implementing novel methods, enabling objective performance assessment. While there are recent benchmarks that focus on code generation in machine learning domain, they do not always require methodological innovation. Works like MLAgentBench [22] and MLE-

Table 2: Comparison between MLRC-BENCH and existing work on automated scientific discovery in machine learning with LLM agents. "∼" means that some but not all of the tasks in that benchmark require the indicated capability. "Compute Constraints" indicates whether the solution code must adhere to specified runtime and GPU memory limitations.

| | Problem Identification | Method Proposal | Experiment Design | Code Implementation | Evaluation Method | Evaluation Object | Compute Constraints | Continual Updates |
|---|---|---|---|---|---|---|---|---|
| AI Scientist [40] | ✓ | ✓ | ✓ | ✓ | LLM & Human Judge | Paper | | |
| Can LLMs Generate Novel Research Ideas? [49] | ✓ | ✓ | ✓ | | Human Judge | Idea Proposal | | |
| DiscoPOP [39] | | ✓ | | ✓ | Performance -Based | Function-Level Code | | |
| MLAgentBench [22] | | ∼ | | ✓ | Performance -Based | Single-Script Code | | |
| MLE-Bench [7] | | ∼ | | ✓ | Performance -Based | Single-Script Code | | |
| MLGym-Bench [43] | | ∼ | | ✓ | Performance -Based | Single-Script Code | | |
| RE-Bench [59] | | ✓ | | ✓ | Performance -Based | Single-Script Code | ✓ | |
| MLRC-BENCH (Ours) | | ✓ | | ✓ | Performance -Based | Repository -Level Code | ✓ | ✓ |

Bench [7] evaluate agents on Kaggle-style ML tasks but prioritize code implementation over novel research contributions. MLE-Bench requires the final submission to be a CSV file, limiting the data modality. Broader benchmarks such as ScienceAgentBench [13] and DiscoveryBench [42] span multiple scientific domains but lack granularity for ML-specific challenges, while CHIME [18] and OpenD5 [15] target auxiliary tasks like literature review or hypothesis generation. DSBench [29] and AAAR-1.0 [38] extend evaluations to data science and general R&D workflows but still fall short of addressing cutting-edge ML research innovation. RE-Bench [59] and MLGym [43] provide ML research task environments but mostly cover outdated or narrow domains (e.g., CIFAR-10 [32]), with six of seven RE-Bench tasks focusing on language modeling. As its tasks are manually curated by experts, RE-Bench is difficult to update and often lags behind emerging research trends. In contrast, MLRC-Bench sources tasks directly from ML conference competitions, ensuring continual inclusion of cutting-edge problems. Moreover, while RE-Bench evaluates single-script solutions, our benchmark features repository-level coding to better reflect real-world research workflows. Besides, existing benchmarks often fail to specify computation constraints (e.g. runtime and GPU memory limit), which are important to encourage efficient yet effective solutions.

Unlike DiscoPOP [39] and DA-Code [23], which focus on function-level programming or data-science workflows, MLRC-BENCH targets repository-level code comprehension and generation, more faithfully reflecting the skills needed to navigate complex research codebases. This setup supports advanced development behaviors, such as generating and editing multiple interdependent files [41] and reusing existing utilities [36]. Consequently, many ML-agent frameworks [26, 53, 37] built for single-file solutions struggle with the richer challenges posed by our benchmark. While general-purpose coding agents like MLAB [22], OpenHands [58], and RepoMaster [56] can handle repository-level tasks and produce valid implementations, they still find it difficult to iteratively optimize algorithmic effectiveness. Moreover, the repository-level design enables multi-agent collaboration, where specialized agents for literature review, idea generation, coding, and evaluation can jointly refine solutions [21, 17, 65, 48, 62].

Automated end-to-end research workflows like The AI Scientist [40, 61] and MLR-Copilot [35] rely largely on subjective reviews of papers or research proposals for evaluating success. In parallel, ResearchAgent [2] iteratively refines ideas through multi-agent feedback, and Chain-of-Idea-Agent [34] organizes literature into progressive chains to stimulate ideation. However, it remains unclear how subjectively evaluated "novel" ideas translate into actual performance gains. In contrast, we explicitly investigate how such subjective assessments of novelty or idea quality align or fail to align with measurable performance improvements. The differences between our benchmark and existing work on automating ML research workflow with LLM agents are presented in Table 2.

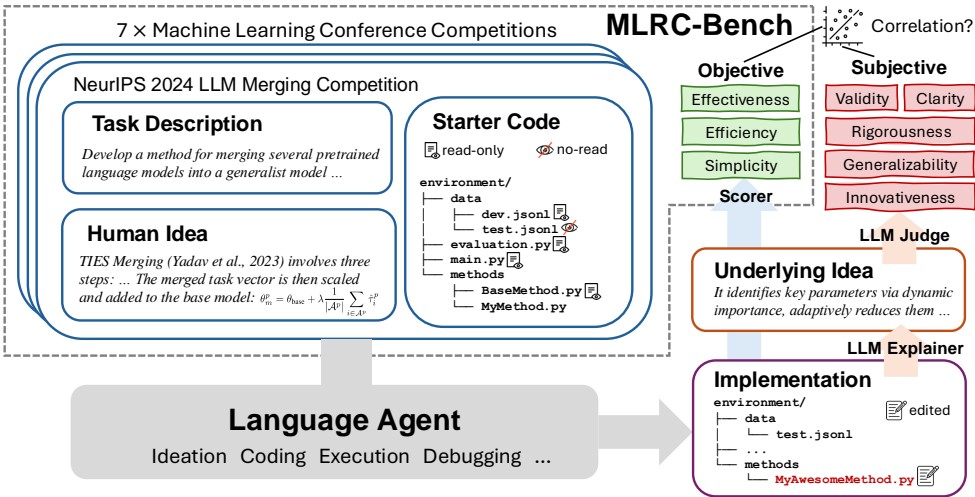

Figure 1: Overview of MLRC-BENCH and its evaluation pipeline. MLRC-BENCH standardizes ML conference competitions into an agent-agnostic framework featuring repository-level code execution under compute constraints. Its evaluation relies on *objective* metrics (effectiveness, efficiency, simplicity) while using subjective LLM-judge scores only to analyze their correlation with objective metrics for assessing LLM-judge reliability (Section 4.3).

## 3 MLRC-BENCH

### 3.1 Task Environment

MLRC-BENCH offers a modular, agent-agnostic environment for specifying and automatically evaluating agents on research tasks. As shown in Figure 1, for each task, we provide:

- **Task Description.** A detailed description of the research problem, including essential terminology, data format, desired model outputs, and constraints (e.g., limitations of model size or training time).

- **Starter Code.** Refactored from official competition repositories, it contains: 1) a simple, baseline model for comparison; 2) a python environment with the necessary ML frameworks/packages; 3) scripts for training, inference, and offline or online evaluation; 4) train, development and test data splits. Training data may not be available for some competitions.

- **Human Idea.** Insights from state-of-the-art papers or top-participant solution reports are included. Agents can optionally utilize these ideas to refine or inspire their own solutions.

**Task-Agonistic Starter Code Structure.** Because our primary goal is to focus on method development, we simplify ML experimentation by refactoring each competition starter kit into a standardized, well-organized format, comparable to common ML research project layouts (Figure 1). The resulting codebase allows users to launch experiments with a single command: `python main.py -method my_awesome_method -phase dev/test`, which applies the specified method to the task and evaluates the result in both development and test phases. To ensure a fair comparison and preserve the integrity of evaluations, the repository enforces file permission management: agents may only modify the `methods/` directory (where the algorithmic logic resides in `MyMethod.py`), while evaluation scripts remain read-only. Additionally, files containing held-out test data are invisible to agents during development phase.

**Development and Test Splits.** We prioritize preventing overfitting by providing explicit development and test splits for each competition. Agents can choose to refine their implementations based on the development set and then submit their best-performing solution to a hidden test set. Wherever possible, we use the original competition test set (via local evaluation or online leaderboard API). Otherwise, we partition the existing development data into custom dev and test sets, reproduce the top human solution if available, and evaluate it on our new test split for a valid comparison.

## 3.2 Task Selection

MLRC-BENCH prizes high-quality competitions that are both non-trivial and reproducible. To form our dataset, we screen competitions held at recent machine learning, natural language processing, data mining, and computer vision conferences or workshops using the following criteria: 1) **Novel Research-Focused:** The tasks should require genuine methodological innovation, rather than being solvable through purely brute-force or superficial engineering approaches, such as exhaustive search for hyperparameters or features without any theoretical motivation or problem understanding. 2) **Non-Trivial:** The problem must involve complexity so that it will not be solved by simply applying standard ML algorithms, e.g., calling the XGBoost classifier [11] on a new dataset or prompt engineering with LLMs. 3) **Feasible:** Starter code, data splits, and evaluation procedures must be publicly available so that researchers, either human or agentic AI, can reproduce the experiments while keeping computational costs manageable.

The current version of MLRC-BENCH comprises seven tasks adapted from existing ML research competitions.[2] These tasks span a diverse landscape of applied ML research, covering topics from LLM safety to multimodal perception (Table 1), and are carefully curated to reflect unsolved, high-impact challenges that demand genuine algorithmic creativity rather than incremental tuning or ensembling. Rather than emphasizing breadth across numerous tasks [7], MLRC-BENCH prioritizes depth and research-grade complexity, where solving even a single task signifies meaningful scientific progress.

In addition, we have the following three considerations for MLRC-BENCH design. **Continual Updates.** MLRC-BENCH evolves with the field by adding new ML conference competitions and retiring tasks where performance has saturated, ensuring alignment with frontier research. We also provide standardized templates and contribution guidelines to encourage community expansion.[3] **Data Contamination Mitigation.** Moreover, since competition solutions are typically shared only as summary reports rather than full implementation code, their content rarely appears in LLM pretraining data, minimizing contamination. Regular updates further ensure that the benchmark evaluates genuine research ability rather than memorized solutions. **Computational Constraints.** Finally, each task enforces explicit runtime and GPU memory limits that mirror real-world competition settings, ensuring fairness and encouraging efficient, resource-conscious methods.

## 3.3 Objective Evaluation Metrics

MLRC-BENCH supports objective evaluation based on three dimensions. We measure **Effectiveness** by the *performance metric* (e.g., accuracy) defined by the competition organizer, **Efficiency** by the solution *runtime* during training (if applicable) and inference, and **Simplicity** in terms of *logical lines of code (LLoC)* [44], inspired by standard practice in software estimation [44]. LLoC excludes comments and blank lines, focusing on executable statements. This metric, while imperfect, offers a rough gauge of code complexity and maintainability [3]. For better readability, we refer to LLoC as "lines of code" throughout this paper.

**Main Metric: Relative Improvement to Human Solution.**  Quantitative performance comparisons across competitions can be tricky, as each task may differ significantly in its intrinsic difficulty, and the official baseline may be weaker or stronger. To address this, we use the *Relative Improvement to Human Solution* as our main leaderboard metric that convert each raw performance score $s_{\text{agent}}$ into a normalized score $s'_{\text{agent}}$, using a linear transformation [5, 59]. Therefore, the score of the baseline solution will be 0, and the top human solution in competition is set to 100. Formally, the normalization is computed as:

$$s'_{\text{agent}} = \frac{s_{\text{agent}} - s_{\text{baseline}}}{s_{\text{top\_human}} - s_{\text{baseline}}} \times 100(\%)$$

---

[2]While numerous competitions have been hosted at recent ML/AI conferences, only a limited subset was selected due to factors such as certain challenges being considered solved with the rapid advancement of foundation models, missing or irreproducible evaluation data/code, qualitative rather than quantitative evaluation, insufficient recency, or limited emphasis on algorithmic innovation.

[3]`https://tinyurl.com/MLRC-Task-Template`

### 3.4 Evaluation Protocol

Our evaluation protocol is designed to prevent AI agents from test set overfitting. Agents will submit their implementation in the form of an edited codebase, particularly within their proposed method in the `methods/` directory. Specifically, in a single trial, an agent can iteratively modify the codebase multiple times. We store snapshots of the codebase immediately after each change. Whenever an execution occurs on the development set, we record the resulting metrics and the name of evaluated method for that snapshot. At the end of this iterative development phase, we pick the snapshot with the best development performance i.e., **Effectiveness**. We then evaluate the method contained in that snapshot on the test set for our final result.[4] This approach strictly follows standard ML practice and ensures reproducible experimentation. Future work may explore more sophisticated multi-objective selection criteria that additionally weigh runtime (**Efficiency**) or lines of code (**Simplicity**) of implementations.

## 4 Experiments and Results

To evaluate the capability of LLM agents in solving ML research tasks, we conduct comprehensive experiments across different agent scaffoldings and language models. Each agent trial is conducted either on a single NVIDIA Quadro RTX 8000 GPU with 48GB of memory (for llm-merging, backdoor-trigger and rainfall-pred tasks) or a Tesla V100 GPU with 16GB memory (for all other tasks), determined by the size of the base model used in each task. Unless otherwise specified, we perform 8 trials[5] per configuration and report the best attempt.

### 4.1 Agent Scaffolding Comparison

In addition to allowing agents to directly propose and implement ideas, we investigate whether providing AI-generated or human-sourced ideas can enhance agent performance. Due to computational cost constraints, we follow the practice of MLE-Bench [7] to evaluate a commonly used model for agentic tasks, GPT-4o [24], under three scaffolding configurations:

- **MLAB**: We adopt the general-purpose MLAB framework [22] as our primary agent for evaluation.[6] MLAB is a ReAct-style [63] agent that alternates between reasoning steps (e.g., reflection, research planning, fact-checking) and actions (e.g., file operations or Python execution) to implement machine learning methods. Notably, MLAB does not include any built-in web search capability; it operates entirely over local resources such as code files, the Python runtime, and its internal memory.

- **CoI-Agent Idea + MLAB**: We augment MLAB with ideas generated by Chain-of-Ideas (CoI) [34], an LLM-based ideation agent that structures relevant literature into progressive reasoning chains. CoI-Agent is equipped with web-based tools, including access to the Semantic Scholar API for retrieving up-to-date research papers. We use OpenAI's o1 [25] model as its backbone to encourage more creative and literature-grounded ideation. By evaluating CoI-Agent-generated ideas with MLAB, we effectively study how agents can leverage web-based retrieval to tackle machine learning research challenges.

- **Human Idea + MLAB**: To test whether agents can achieve stronger performance when given high-quality conceptual guidance, we provide MLAB with human-curated ideas manually extracted from state-of-the-art papers or top-performing participants' reports.

For all tasks, MLAB agents are limited to 50 steps and 5 hours per trial, except for Rainfall Prediction, which allows 100 steps and 10 hours to match the longer training time of the official baseline. This ensures fair comparison, as the baseline requires more epochs to converge due to a larger dataset. As shown in Table 3, incorporating additional ideas, whether generated by AI or proposed by humans, does not consistently improve performance. This underscores *the challenges agents face not only in*

---

[4]Concretely, we execute the command `python main.py -method best_dev_method -phase test`.

[5]The number of trials is limited to 8 due to budget constraints on API usage.

[6]While existing ML agent frameworks, including AIDE [26], SELA [14], and AutoML-Agent [55], demonstrate strong performance on Kaggle-style tasks that yield single-file solutions, their design assumptions differ substantially from our repository-level coding setup, necessitating careful adaptation for effective use.

Table 3: For each competition and agent, we report the test-phase *relative improvement to the human solution*. Best performing agent in each task is highlighted in **bold**. Our results indicate that providing additional ideas, whether sourced from AI or humans, does not consistently yield performance improvements. The best-performing configuration, gemini-exp-1206 under MLAB, achieves only 9.3% of the human-level improvement over baseline on average, underscoring the inherent difficulty of these research tasks. See Table 4 in Appendix B for *absolute improvements to baseline solution*.

| Agent | temporal -action-loc | llm -merging | meta -learning | product -rec | rainfall -pred | machine -unlearning | backdoor -trigger | Avg |
|---|---|---|---|---|---|---|---|---|
| MLAB (gemini-exp-1206) | -0.5 | **5.0** | -1.1 | 0.1 | 43.1 | 5.6 | 12.9 | **9.3** |
| MLAB (llama3-1-405b-instruct) | 0.5 | -1.0 | -4.9 | 0.0 | 31.5 | 6.2 | 11.5 | 6.3 |
| MLAB (o3-mini) | 0.3 | -1.0 | -4.9 | 0.1 | 25.1 | 3.6 | 6.2 | 4.2 |
| MLAB (claude-3-5-sonnet-v2) | **0.8** | **5.0** | -4.9 | **3.0** | 14.6 | -94.7 | **39.9** | -5.2 |
| MLAB (gpt-4o) | 0.3 | 2.0 | -4.9 | 0.6 | **47.5** | -18.0 | 10.4 | 5.4 |
| w/ Human Idea | 0.5 | -1.0 | -4.9 | 2.2 | 12.3 | 6.8 | 8.8 | 3.5 |
| w/ CoI-Agent Idea (o1) | 0.4 | -1.0 | -4.9 | 0.1 | 39.4 | **11.8** | 4.0 | 7.1 |

*generating high-quality ideas but also in effectively implementing them, even when the ideas originate from humans.*

## 4.2 Model Comparison

Taking MLAB as our major scaffold, we evaluate five prominent LLMs: Claude 3.5 Sonnet v2 [1], gemini-exp-1206 [46][7], Llama 3.1 405B Instruct [16], o3-mini-high [45] and GPT-4o (2024-11-20) [24]. The results in Table 3 show varying success rates across models and tasks. Among all models, Gemini-exp-1206 performs the best overall, closing 9.3% of the gap between baseline and top human participant scores. Claude 3.5 Sonnet V2 performs the best on most tasks but fails significantly on machine unlearning. A case study of its final solution (Appendix C.1) suggests that the failure stems from treating the removal of unwanted data and the preservation of useful knowledge as separate objectives, rather than jointly optimizing both to maintain model performance.

Table 3 also reveals that agent solutions' performance gains remain modest compared to human solutions in many cases, if not degrading baseline performance. There are, however, a few notable exceptions. For instance, MLAB (gpt-4o) achieves a score of 47.5 on the rainfall prediction task, likely because similar solutions (e.g., variants of U-Net [47]) are readily available online. In the backdoor-trigger task, the baseline GCG method [68] performs poorly, where it essentially makes random predictions, thereby lowering the bar for agents to surpass it with more meaningful solutions. This substantial gap highlights *the current limitations of AI agents in generating novel, effective methods*, underscoring the need for further advances to match or surpass human-led research efforts.

## 4.3 Reliability of Subjective Evaluations with LLM-as-a-Judge

Our benchmark enables an investigation of whether LLM-as-a-judge evaluations [33, 66] can reliably assess the quality of research ideas by comparing subjective judgments against objective performance. As illustrated in Figure 1, we first prompt an LLM to explain each implementation's underlying idea.[8] Following a peer-reviewed rubric [2], an LLM then assigns 1–5 Likert scores for the explained ideas on five dimensions: `validity`, `clarity`, `rigorousness`, `generalizability`, and `innovativeness`. These scores are all the higher the better. The prompts used for this evaluation are shown in Appendix E. We adopt OpenAI's o1 [25] model as both the idea explainer and the judge for its strong reasoning and coding capabilities. The subjective scores serve only to analyze potential biases in LLM-based evaluation; they are *not* part of the benchmark evaluation protocol.

Furthermore, we examine how the presence of code influences the assessments through two settings. (1) *Without Code*, in which the LLM judges only access the task description and the proposed idea; and (2) *With Code*, in which the judges also see the code implementation. We then compute Spearman's correlation [51] for each pair of objective and subjective metrics, using data from all valid implementations that include test-phase scores.

---

[7]Gemini-exp-1206 is renamed to Gemini 2.0 Pro afterwards.

[8]We prompt the MLAB agent to include detailed comments in the code to enable faithful explanations.

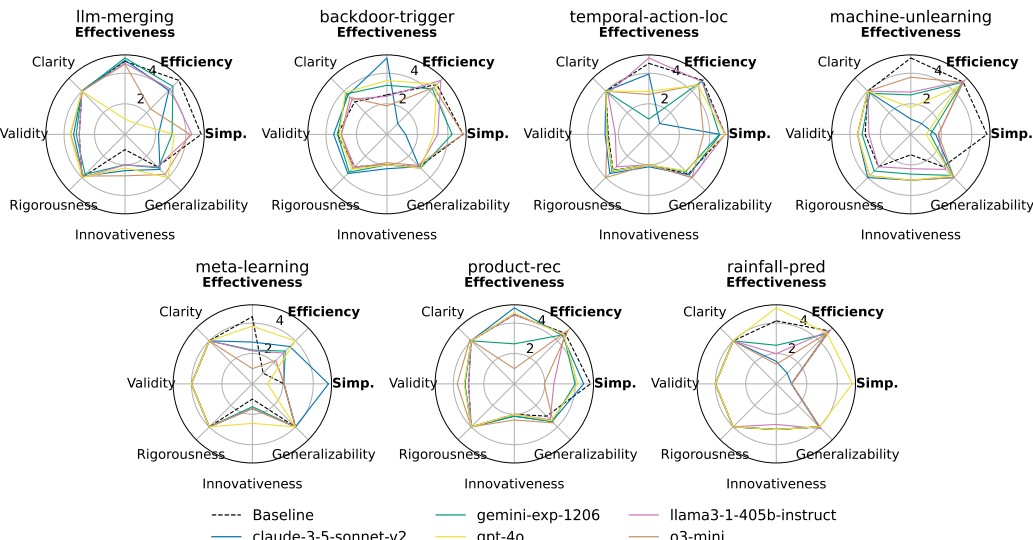

Figure 2: Radar plots of objective and subjective evaluations for agent-generated solutions across seven research tasks. Each dimension is normalized on a 1–5 scale, where higher values indicate better performance. *Objective* metrics include **Effectiveness**, **Efficiency**, and **Simplicity (Simp.)**, which are highlighted in **bold**. The rest are *subjective* metrics, assessed by prompting o1 as a judge. Notably, more effective solutions identified by agents tend to be more complex and time-consuming (e.g., in backdoor trigger recovery). Additionally, overlapping scores in subjective dimensions suggest that LLM-based evaluation struggles to distinguish the research capabilities of different models.

Figure 2's radar plots provide a holistic view of agent performance across both subjective and objective dimensions. The plots show that while agents occasionally produce effective solutions, they often struggle to balance other criteria such as efficiency and simplicity. For instance, on the backdoor-trigger task, Claude 3.5 Sonnet V2 scores well on effectiveness but poorly on efficiency and simplicity, suggesting that agent-generated solutions tend to be more complex and time-consuming. Notably, agents generally underperform compared to the baseline when evaluated using objective metrics. However, when subjective metrics are used and judged by LLMs, they often receive more favorable ratings. This discrepancy highlights *a risk of overly optimistic conclusions when relying solely on subjective evaluations*.

Figures 3 and 7 (in Appendix B) illustrate the correlation heatmaps for both settings.[9] The overall correlations remain weak. For example, there is a near-zero correlation (-0.06) between innovativeness

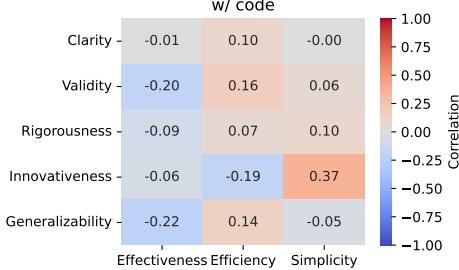

Figure 3: Correlation heatmap between objective (x-axis) and subjective (y-axis) metrics for agent-generated solutions across all tasks. Code is included when prompting the LLM to evaluate subjective dimensions. No strong correlation is observed, suggesting that LLM-judged subjective metrics may not reliably indicate empirical performance gains.

and effectiveness, implying that an agent's ability to generate novel ideas, as judged by an LLM, does not necessarily equate to success in practical tasks. Consequently, our finding indicates that *LLM-based evaluations alone are not a reliable proxy for real-world research impact*. While LLM agents can certainly assist in generating creative ideas, relying solely on LLM-based evaluations to gauge agents progress in improving machine learning research may lead to misinterpretations. This again highlights the importance of employing objective metrics to ensure that proposed solutions are not only novel but also effective.

---

[9]We find that removing the code leads to similar correlation results and does not significantly affect the conclusion we make.

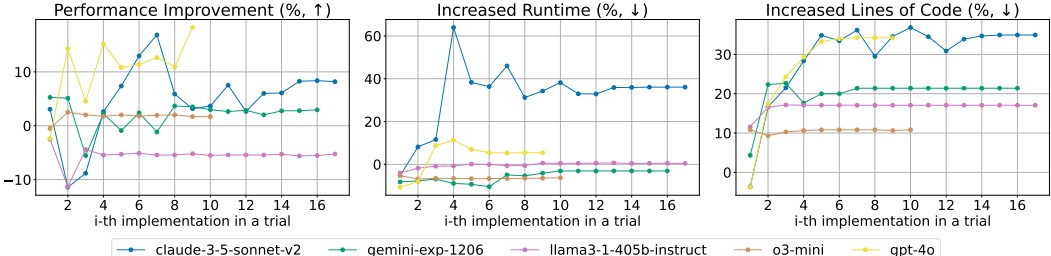

Figure 4: We track the percentages of changes of performance, runtime, and lines of code compared to baseline across iterative refinement of implementations within a trial of LLM-based MLAB agent on the development set. Performance improvement is the higher the better, while increased runtime and lines of code are the lower the better. These figures show the averaged metrics across all tasks. For results breakdown on each task, please refer to Figure 9 and 10 in Appendix B. Together, these figures show that agents tend to over-refine their solutions over time, leading to increased complexity and runtime without proportional performance gains.

## 4.4 Solution Development Analysis

Figure 4 illustrates how performance, runtime, and code complexity evolve as agents iteratively refine their implementations within a single trial. Three major patterns are observed: (1) GPT-4o and Claude gradually improve their performance through refinement, while other models plateau after a few iterations; (2) runtime consistently increases, probably because models are exploring more complicated solutions over time, which may naturally conflate with better solutions; and (3) code size expands over time, reflecting increasingly complex solutions that do not yield proportional performance gains. Together, these trends suggest that *agents tend to over-refine their solutions, resulting in more complex and time-consuming implementations without further performance improvements*.

We further analyze the agent traces (gemini-exp-1206 with MLAB) in Appendix F to understand its behavioral patterns and limitations. Two key insights emerge. First, a significant portion of action errors (11.5% of all steps) stem from incorrect tool arguments, where the model either hallucinates or misidentifies expected parameter names. This highlights the need to *improve the agent's tool-usage capabilities*. Second, the agent was able to fix only 17.2% of the errors encountered during code execution, revealing the *challenge of self-debugging in complex, repository-level ML codebases* [12].

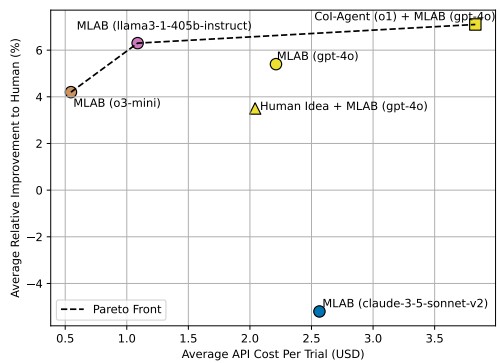

Figure 5: We perform a cost-effectiveness analysis of various setups. On the x-axis, we plot API cost, where lower is better, and on the y-axis, we show relative improvement to human (Section 3.3), where higher is better.

## 4.5 Cost-Effectiveness Analysis

In Figure 5, we analyze the agents' success rates in a cost-controlled setting, motivated by recent work [30] emphasizing the importance of jointly optimizing both performance and cost in agent design.[10] Llama 3.1 405b Instruct[11] offers the most favorable trade-off, achieving higher success rates than GPT-4o and Claude 3.5 Sonnet at a significantly lower cost. Although incorporating an ideation phase before implementation improves overall performance compared to the implementation-only MLAB setting, it incurs additional costs due to the generation of research ideas. Nevertheless, we believe the performance gain will

---

[10]We exclude the `gemini-exp-1206` model from this figure because it was experimental and its pricing was unavailable at the time of writing.

[11]We estimate the API cost for Llama models based on Amazon Bedrock service pricing.

increasingly justify the added cost as base models continue to grow stronger, particularly for complex research problems where strategic high-level planning leads to substantial gains in final outcomes.

### 4.6 Inference-Time Scaling on ML Research Tasks

Increasing inference-time compute via repeated sampling [7, 8, 4] has been shown to boost LLM performance on reasoning and coding tasks. Here we explore how LLM research agents scale with more inference-time compute on both the idea and solution spaces. We sample 4 ideas for each task from CoI-Agent [34] and repeat MLAB agent for 8 trials to implement each idea into code.

Figure 6 plots pass@$k$ [9], i.e., the probability that at least one of $k$ trials converges to a successful implementation, defined as the agent closes at least 5% of the gap between baseline and top human participant scores (Relative Improvement to Human, Section 3.3). Our results show that *providing high-quality ideas enhances an agent's ability to generate meaningful solutions when given multiple attempts*, and *human ideas appear to be more effective than those produced by AI*.[12] Furthermore, under a fixed inference budget, we did not observe a significant difference between allocating resources to idea exploration versus exploitation. For example, there is no significant pass@k difference between using 4 ideas with 2 trials per idea, 2 ideas with 4 trials per idea, and 1 idea with 8 trials per idea. This phenomenon likely occurs because once a high-quality idea is identified, the performance gains from additional trials tend to plateau, resulting in diminishing returns despite further exploitation. We hypothesize that performance-informed tree-search that navigates the

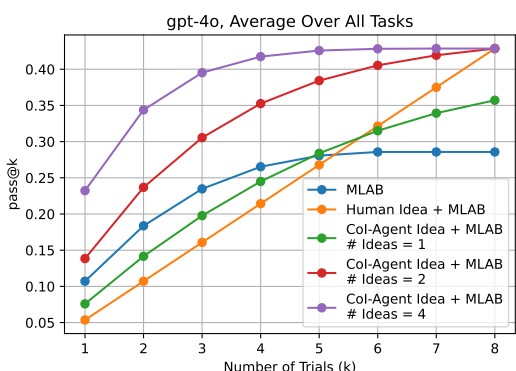

Figure 6: We measure Pass@$k$ as we scale the number of trials and ideas, running MLAB for eight trials per idea. The total inference-time computes are equivalent among these points: $k = 4$ for one-idea line, $k = 2$ for two-idea line, $k = 1$ for four-idea line, and $k = 4$ for the remaining lines. For results breakdown on each task, please refer to Figure 8 in Appendix B. Our results indicate that 1) providing high-quality ideas, especially human-generated ones, significantly boosts an agent's success rate across multiple attempts, 2) while varying the balance between idea exploration and exploitation under a fixed budget yields similar outcomes due to diminishing returns from repeated trials.

vast space of possible solutions [26, 31] or allocating more computational resources [7] could offer more promising scaling properties.

## 5 Conclusion

MLRC-BENCH draws upon the rigor of conference competitions to provide a scalable, objective, and realistic benchmark for evaluating LLM agents in proposing and implementing novel algorithms that advance research on impactful topics. Our benchmark features modular tasks, objective evaluation metrics, tamper-proof evaluations, and ongoing updates as new suitable competitions become available. Our results show that MLRC-BENCH presents a significant challenge for state-of-the-art LLMs and agent scaffoldings. Our analysis highlights the misalignment between the LLM-judged innovation and their actual performance on cutting-edge ML research problems. MLRC-BENCH will evolve alongside the rapid pace of ML research and continuously support the pursuit of AI-assisted or automated scientific discovery.

---

[12]As shown in Figure 8 of Appendix B, most tasks exhibit zero successes across all eight trials, suggesting that further increasing the number of trials would not alter their Pass@k curves. To examine this more closely, Appendix C.2 extends the scaling experiments to 16 trials on two representative tasks—backdoor-trigger-recovery and machine-unlearning—revealing consistent conclusions.

## Acknowledgments

This work is supported by LG AI Research and computational resources and services provided by Advanced Research Computing (ARC), a division of Information and Technology Services (ITS) at the Unversity of Michigan, Ann Arbor. We thank Xinnuo Li, Aryan Sharma, Jett Rosen, and Sandeep Jala for their help with the survey and collection of research competitions. We also thank LAUNCH lab members for their useful feedback.

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

# A  Detailed Description of Research Competitions

## A.1  LLM Merging [52] [Link]

Develop a novel and effective LLM merging method to improve performance on held out test set within the time constraints.

## Description
Training high-performing large language models (LLMs) from scratch is a notoriously expensive and difficult task, costing hundreds of millions of dollars in compute alone. These pretrained LLMs, however, can cheaply and easily be adapted to new tasks via fine-tuning, leading to a proliferation of models that suit specific use cases. Recent work has shown that specialized fine-tuned models can be rapidly merged to combine capabilities and generalize to new skills.

The competition will provide the participants with a list of expert models that have already been trained on a task-specific dataset. The goal of this competition is to re-use the provided models to create a generalist model that can perform well on a wide variety of skills like reasoning, coding, maths, chat, and tool use. Along with these expert models, we have a set of hidden tasks that will be used to evaluate the submissions from participants.

## A.2  Backdoor Trigger Recovery [60] [Link]

**Backdoor Trigger Recovery for Code Generation Models**

## Description

Participants in this competition are tasked with developing algorithms to recover backdoor triggers embedded within large language models (LLMs) used for code generation. Each provided backdoored LLM contains multiple (trigger, target) pairs, where triggers are universal prompt injections designed to induce the generation of malicious code specified by the targets. In the development phase, participants receive a model finetuned with five known (trigger, target) pairs, while in the testing phase, the models include tens of secret (trigger, target) pairs related to various categories of harmful code generation. The objective is to predict the triggers corresponding to each provided target, adhering to a maximum token constraint of 10 tokens per trigger. Submissions will be evaluated using two metrics: recall, which measures the similarity between predicted and ground truth triggers, and the Reverse-Engineering Attack Success Rate (REASR), which assesses the effectiveness of the recovered triggers in eliciting the malicious code. Participants are provided with a starter dataset of 100 code generation queries and their correct outputs for method development and local evaluation, with additional data encouraged for enhancing method robustness. However, any attempts to access or guess the secret online evaluation dataset will be considered a rule violation.

## A.3  Temporal Action Localisation [20] [Link]

# Second Perception Test Challenge (ECCV 2024 Workshop) – Temporal Action Localisation Track

## Description
The goal of this challenge is to develop methods that accurately **localize and classify actions** in untrimmed videos (up to 35 seconds long, 30 fps, max resolution 1080p) from a predefined set of classes.

—

## Data
- **Training Data: Multimodal List**
- 1608 videos
- Includes both **action** and **sound** annotations
- Contains **video and audio features**

- **Validation Set**
- 401 videos, used to tune hyperparameters.

- **Test Set**
- Held-out set for final evaluation of your method's performance containing 5359 videos.

—

## Output Format
For each video in test (or val), your model should output **all action segments**, with:
1. **Start timestamp**
2. **End timestamp**
3. **Predicted action class label**
4. **Confidence score**

—

## Evaluation
- The main metric is Mean Average Precision (mAP), computed over your detected segments and averaged across:
- Different action classes
- IoU thresholds from 0.1 to 0.5 in increments of 0.1 (i.e., [0.1, 0.2, 0.3, 0.4, 0.5])
- You have separate splits for train, val, and test:
- Train on the training set.
- Use the validation set to tune, select models, etc.
- Evaluate final performance on the **test set**.

### A.4    Rainfall Prediction [19] [Link]

Super-Resolution Rain Movie Prediction under Temporal Shifts

## Description
The aim of the Weather4cast competition is to predict quantitatively future high resolution rainfall events from lower resolution satellite radiances. Ground-radar reflectivity measurements are used to calculate pan-European composite rainfall rates by the Operational Program for Exchange of Weather Radar Information (OPERA) radar network. While these are more precise, accurate, and of higher resolution than satellite data, they are expensive to obtain and not available in many parts of the world. We thus want to learn how to predict this high value rain rates from radiation measured by geostationary satellites operated by the European Organisation for the Exploitation of Meteorological Satellites (EUMETSAT).

Competition participants should predict the exact amount of rainfall for the next 8 hours in 32 time slots from an input sequence of 4 time slots of the preceeding hour. The input sequence consists of four 11-band spectral satellite images. These 11 channels show slightly noisy satellite radiances covering so-called visible (VIS), water vapor (WV), and infrared (IR) bands. Each satellite image covers a 15 minute period and its pixels correspond to a spatial area of about 12km x 12km. The prediction output is a sequence of 32 images representing rain rates from ground-radar reflectivities. Output images also have a temporal resolution of 15 minutes but have higher spatial resolution, with each pixel corresponding to a spatial area of about 2km x 2km. So in addition to predicting the weather in the future, converting satellite inputs to ground-radar outputs, this adds a super-resolution task due to the coarser spatial resolution of the satellite data.

We provide training and validation data from one Eureopean region in 2019, and testing data from the same region in 2020, measuring a transfer learning performance under temporal shift. The task is to predict exact amount of rain events 4 hours into the future from a 1 hour sequence of satellite images. Rain rates computed from OPERA ground-radar reflectivities provide a ground truth.

### A.5    Machine Unlearning [54] [Link]

# Machine Unlearning Challenge

**One-sentence summary**
Develop efficient algorithms for "machine unlearning" such that, after forgetting certain training data, the resulting model closely matches one that was never trained on that data in the first place.

—

## Description

We focus on **machine unlearning**, i.e., "removing the influence" of a subset of the training data (the *forget set*) from a trained model, so that the resulting model behaves similarly to one trained *without* that subset. This is especially relevant for privacy regulations (e.g., "right to be forgotten"), where individuals can request removal of their data from a model.

### Goal

Our goal is to compare the strengths and weaknesses of different unlearning methods under a *shared* and *standardized* evaluation. Participants receive:

1. A **pre-trained** model (trained on facial images, CASIA-SURF, to predict age group in test phase, CIFAR-10 in dev phase).
2. A **forget set** (data samples to remove) and a **retain set** (the rest of training data).
3. A hidden **test set** for final scoring.

**Output**: An unlearned model that should:
- **Erase** the forget set's influence to match the behavior of a retrained model that never saw those forget samples.
- **Retain** good accuracy on the remaining data and on the test set.
- **Finish** within provided compute/runtime constraints.

### Data & Evaluation

- **Dataset**: CASIA-SURF, containing facial images labeled by age group (10 classes) in test phase, CIFAR-10 in dev phase.
- **Pretrained model**: A classifier trained for 30 epochs on the entire dataset.
- **Forgetting**: Must "remove" any trace of the forget set.
- **Utility**: Must stay accurate on the retain data and a hidden test set.
- **Metrics**:
1. **Forgetting quality** – compares unlearned model $\theta_u$ to a model retrained from scratch $\theta_r$ without the forget set.
2. **Utility** – checks retain/test accuracy relative to $\theta_r$.
3. **Efficiency** – run under time constraints ($< 8$h on provided compute).

The challenge uses an *online* evaluation on Kaggle. Each submitted unlearning method will be run multiple times against multiple "original" and "retrained-from-scratch" checkpoints, producing a final score that balances forgetting quality and model utility.

### A.6   Next Product Recommendation [28] [Link]

This task focuses on next product recommendation by predicting the most likely product a customer will engage with based on session data and product attributes, using test data from English, German, and Japanese locales.

## Description
For each session, the participant should predict 100 product IDs (ASINs) that are most likely to be engaged with. The product IDs should be stored in a list and are listed in decreasing order of confidence, with the most confident prediction at index 0 and least confident prediction at index 99. Evaluation is performed using mean reciprocal rank where the rank in your list of the ground truth next item is being assessed. For each session, you will be provided with the locale of the user and a list of products already viewed in that session. A separate file has metadata about each product.

### A.7 Cross-Domain Meta Learning [6] [Link]

The competition focuses on cross-domain meta-learning for few-shot image classification, challenging participants to develop scalable and robust models that can quickly adapt to diverse tasks with varying numbers of classes ("ways") and training examples per class ("shots") across domains like healthcare, ecology, and manufacturing.

## Description
Goal and Data
This competition challenges participants to develop meta-learning models that adapt quickly to few-shot classification tasks across ten diverse domains (e.g., healthcare, ecology, manufacturing). Drawing on the newly expanded Meta Album meta-dataset (10 image datasets unified at 128×128 resolution), the final evaluation tasks vary in "ways" (2–20 classes) and "shots" (1–20 training examples per class). By combining such heterogeneous tasks, the challenge highlights the importance of scalability, robustness to domain shifts, and flexible generalization in the "any-way any-shot" meta-learning setting. 5 datasets will be used for training and 5 will be used for testing.
Participants develop a 'MetaLearner' whose 'meta_fit' function returns a 'Learner' whose 'fit' function returns a 'Predictor' with a 'predict' function.

Evaluation and Metric
Submissions are evaluated with blind testing on ten representative datasets. Each task includes a support set (training) and a query set (testing), and the competition's primary metric is a random-guess normalized balanced accuracy. First, a balanced classification accuracy (bac) is computed by averaging per-class accuracies (i.e., macro-average recall). Then, to account for varying numbers of classes (ways), the bac is normalized by the expected performance of random guessing. This ensures a fair comparison across tasks with different ways/shots configurations and highlights each model's true ability to learn effectively from limited examples in multiple domains.

## B  Additional Results

This section presents additional results that complement the findings reported in the main paper and appendix.

- Table 4 reports the absolute improvement over the baseline, supplementing the success rate results shown in Table 3 (Section 4.2).

- Figure 7 displays the correlation heatmap between objective and subjective metrics when LLM-as-a-Judge is applied without code as input, complementing Figure 3 (Section 4.3).

- Figure 8 shows inference-time scaling results broken down by task, complementing the aggregate results in Figure 6 (Appendix 4.6).

- Figures 9 and 10 provide a task-level analysis of the implementation process, extending the results in Figure 4 (Appendix 4.4).

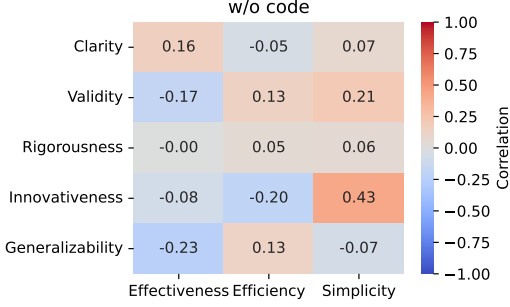

Figure 7: Correlation heatmap between objective and subjective metrics when LLM-as-a-Judge is done without code as input. The "with code" version is shown in Figure 3.

## C  Additional Analyses

### C.1  Case Study

We present two case studies below to illustrate failed solutions implemented by LLM agents. Please see Appendix G for the concrete code implemented by AI agents.

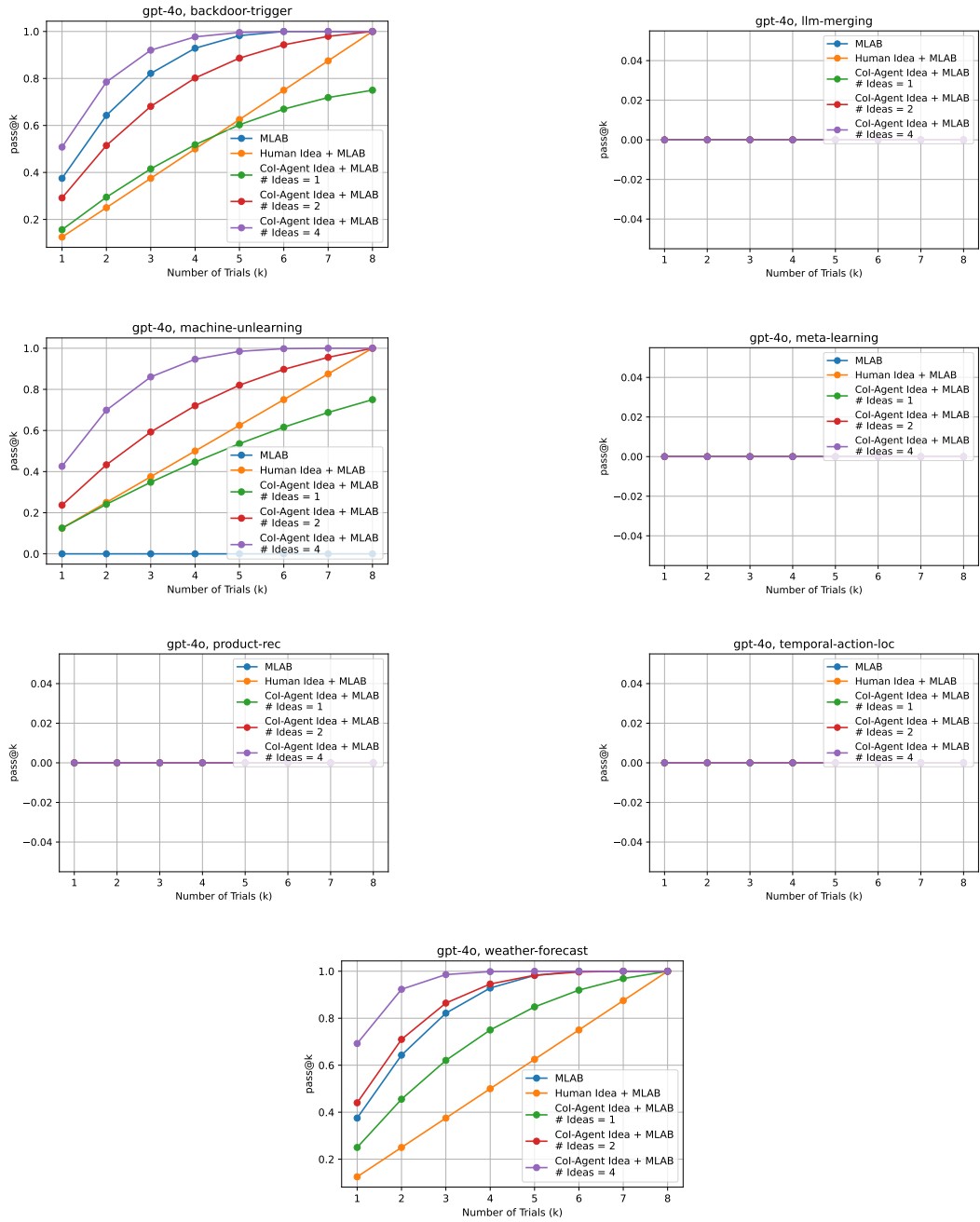

Figure 8: For each task, we measure Pass@$k$ as we scale the number of trials and ideas, running MLAB for eight trials per idea. Pass@$k$ is the probability that at least one of k trials converges to a successful implementation, defined as the agent closes at least 5% of the gap between baseline and top human participant scores.

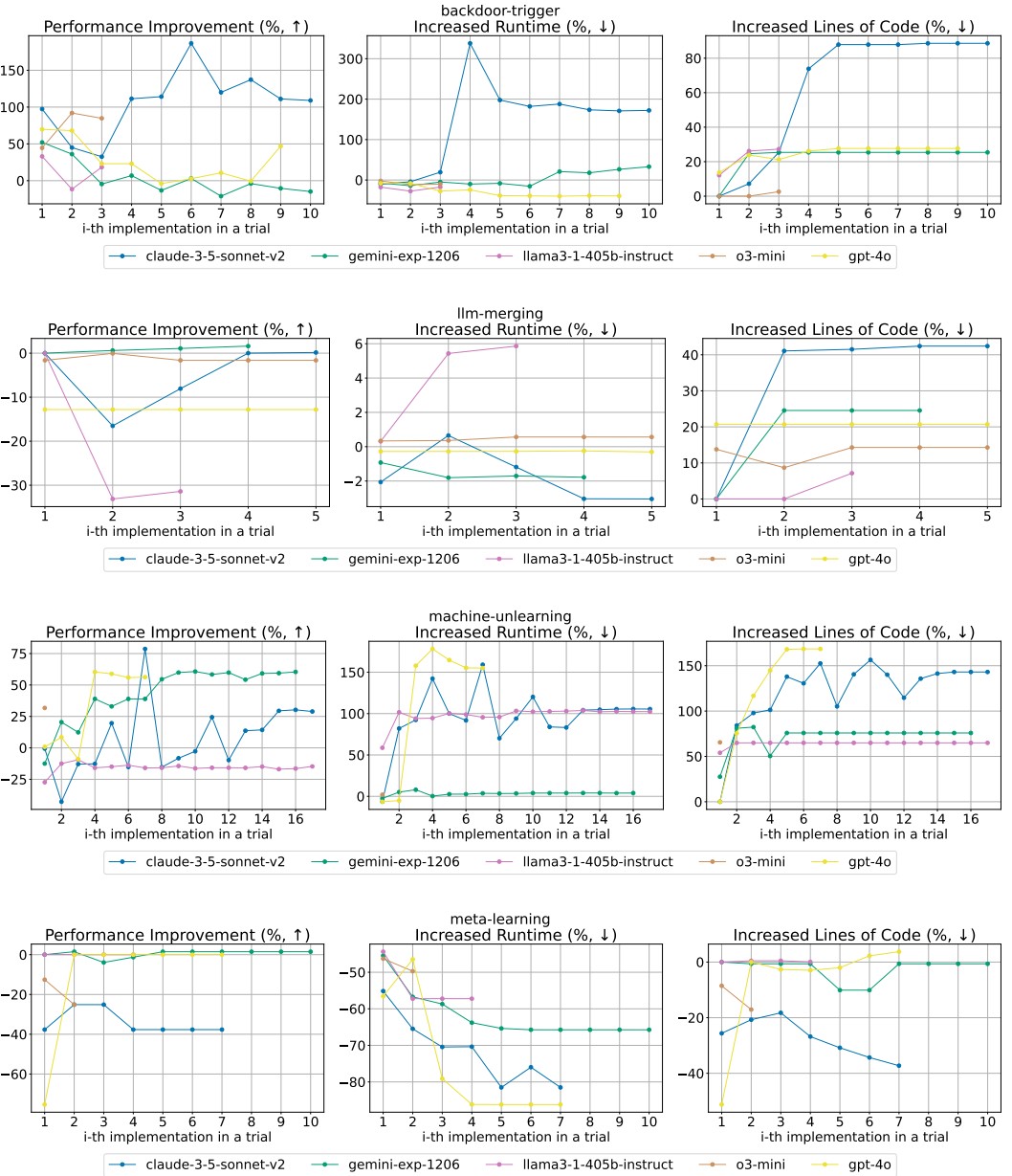

Figure 9: For each task, we track the percentages of changes of performance, runtime, and lines of code compared to baseline across iterative refinement of implementations within a trial of LLM-based MLAB agent.

Table 4: For each research competition and agent, we report the test-phase *best percentage improvement in the performance metric over the baseline among 8 trails* provided in the starter code. Additionally, we present the improvements achieved by the top human participants at the time of competition under the same setup. Best performing agent in each task is highlighted in **bold**. Agents can only achieve marginal performance gains compared to human experts, and in many cases, the agents' solutions even degrade baseline performance.

| Agent | temporal -action-loc | llm -merging | meta -learning | product -rec | rainfall -pred | machine -unlearning | backdoor -trigger | Avg |
|---|---|---|---|---|---|---|---|---|
| MLAB (gemini-exp-1206) | -1.3 | **3.4** | -3.2 | 0.6 | 91.4 | 3.5 | 80.4 | 25.0 |
| MLAB (llama3-1-405b-instruct) | 1.5 | -0.7 | -14.9 | 0.0 | 66.7 | 3.8 | 71.7 | 18.3 |
| MLAB (o3-mini) | 0.9 | -0.7 | -14.9 | 0.6 | 53.3 | 2.2 | 38.8 | 11.5 |
| MLAB (claude-3-5-sonnet-v2) | **2.2** | **3.4** | -14.9 | **12.3** | 31.0 | -58.6 | **247.9** | **31.9** |
| MLAB (gpt-4o) | 0.9 | 1.4 | -14.9 | 2.6 | **100.8** | -11.1 | 64.5 | 20.6 |
| w/ Human Idea | 1.5 | -0.7 | -14.9 | 8.9 | 26.1 | 4.2 | 54.5 | 11.4 |
| w/ CoI-Agent Idea (o1) | 1.0 | -0.7 | -14.9 | 0.6 | 83.6 | **7.3** | 24.9 | 14.5 |
| Top Human in Competition | 284.6 | 68.2 | 304.5 | 412.6 | 212.0 | 61.9 | 621.3 | 280.7 |

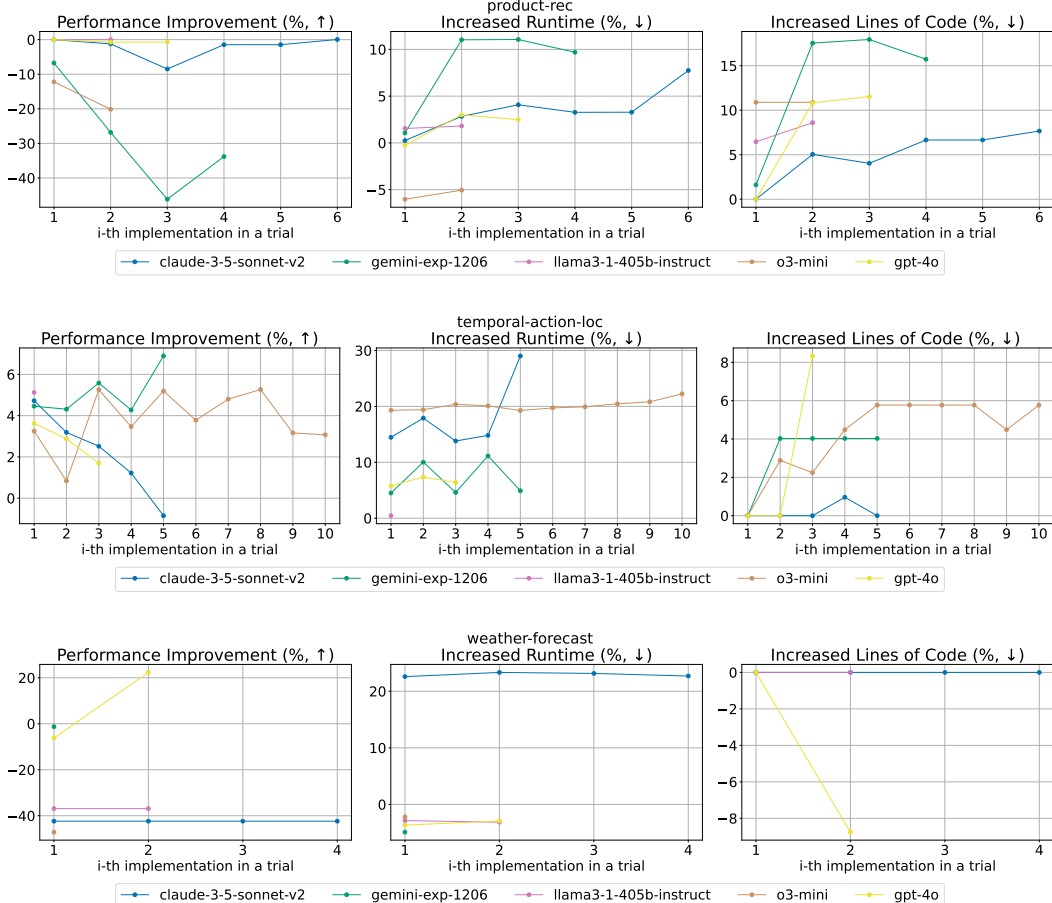

Figure 10: (Cont'd) For each task, we track the percentages of changes of performance, runtime, and lines of code compared to baseline across iterative refinement of implementations within a trial of LLM-based MLAB agent.

Table 5: Extended Pass@k results (up to 16 trials) on two representative tasks, backdoor-trigger-recovery and machine-unlearning, across three system configurations (MLAB-only, CoI-Agent Idea + MLAB, and Human Idea + MLAB). Consistent with the conclusion from Figure 6 in the main paper, incorporating ideation, whether AI- or human-generated, substantially improves success rates over the implementation-only baseline, with human ideas enabling faster and more consistent solution discovery.

| Task | System | pass@1 | pass@4 | pass@8 | pass@12 | pass@16 |
|---|---|---|---|---|---|---|
| backdoor-trigger-recovery | MLAB | 0.12 | 0.45 | 0.77 | 0.95 | 1.00 |
| | CoI-Agent Idea + MLAB | 0.19 | 0.61 | 0.90 | 0.99 | 1.00 |
| | Human Idea + MLAB | **0.31** | **0.82** | **1.00** | **1.00** | 1.00 |
| machine-unlearning | MLAB | 0.00 | 0.00 | 0.00 | 0.00 | 0.00 |
| | CoI-Agent Idea + MLAB | 0.06 | 0.25 | 0.50 | 0.75 | 1.00 |
| | Human Idea + MLAB | **0.12** | **0.45** | **0.77** | **0.95** | 1.00 |

**LLM Merging Challenge:** The objective is to develop a novel and effective algorithm to merge several (in our case, two) expert models into a single model that demonstrates improved performance on a held-out test set within the given time constraints. The MLAB agent (o3-mini) proposed a median aggregation of parameters, which slightly underperforms the baseline of a mean aggregation. We hypothesize that the median, while robust to extreme outliers, typically exhibits higher statistical variability when merging multiple parameter sets, especially with fewer models.

**Machine Unlearning Challenge:** The goal is to develop efficient algorithms that enable a model to "forget" specific training data, such that the resulting model closely resembles one that was never trained on that data in the first place. The MLAB agent (claude-3-5-sonnet-v2) proposed a Gradient Ascent Unlearning Algorithm, a two-phase approach combining gradient ascent for forgetting and fine-tuning for retaining knowledge. Specifically, the algorithm first performs gradient ascent on the forget set to maximize loss (achieving unlearning) and then fine-tunes the model on the retain set to restore the desired knowledge. While this approach sounds promising in theory, it scored significantly lower than the baseline. We hypothesize that by separating the gradient ascent on the forget set and the fine-tuning on the retain set into two distinct phases, the model may not effectively balance these two conflicting objectives. In contrast, a joint optimization approach—where both objectives are optimized at each gradient update—might better balance the processes of "forgetting" and "retaining" knowledge.

### C.2 Extended Inference-time Scaling Experiment

In Section 4.6, we conduct inference-time scaling evaluation with 8 trials at most. As shown in Figure 8, most tasks record zero successes across all eight trials, so we expect that increasing trials would not change their Pass@k curves. To probe further, we extended two representative tasks, backdoor-trigger-recovery and machine-unlearning, to 16 trials for all three settings: MLAB-only, CoI-Agent Idea + MLAB, and Human Idea + MLAB. Results in Table 5 underscore the importance of ideation: on both tasks, providing ideas, either from AI or human, consistently improves Pass@k compared to the MLAB-only (implementation-only) setting, with human ideas enabling faster solution discovery than AI-generated ones. This reinforces our conclusion in Section 4.6 that without true innovation capabilities, simply increasing the number of trials is unlikely to yield better performance.

## D   Limitations

Our work has certain limitations. First, the benchmark currently covers only seven competitions, though we plan to expand it with new tasks as outlined in Section 3.2. Second, our evaluation focuses solely on a general-purpose MLAB agent for code implementation, as some other agent frameworks are currently unsuitable for our repository-level coding tasks (see Section 4.1). Nonetheless, recent advancements in LLM agents, including improved debugging tools [64], inference-time scaling techniques [57, 31], and reinforcement learning-based fine-tuning [67, 27, 10, 50], have the potential to achieve substantially better performance at a lower cost than human researchers.

# E  Prompts for LLM-as-a-Judge

---

**Prompt for LLM Explainer in Figure 1**

Analyze the following Python code with comments and generate a high-level idea proposal summarizing:
1. The main goal or purpose of the method or algorithm implemented.
2. The general approach or methodology used to achieve the goal.
3. Any core assumptions or requirements underlying the implementation.
Focus on providing a conceptual overview rather than implementation details.

Code:
{code}

Provide the summary as an idea proposal, avoiding references to the code itself. Focus on describing the approach and methodology as a standalone concept.

---

**Prompt for LLM Judge in Figure 1**

You are an AI assistant whose primary goal is to assess the quality and soundness of scientific methods across diverse dimensions, in order to aid researchers in refining their methods based on your evaluations and feedback, thereby enhancing the impact and reach of their work.
You are going to evaluate a scientific method for its {metric} in addressing a research problem, focusing on how well it is described in a clear, precise, and understandable manner that allows for replication and comprehension of the approach.
As part of your evaluation, you can refer to the research problem, which will help in understanding the context of the proposed method for a more comprehensive assessment.

Research problem: {researchProblem}

Now, proceed with your {metric} evaluation approach that should be systematic:
- Start by thoroughly reading the proposed method and its rationale, keeping in mind the context provided by the research problem, and existing studies mentioned above.
- Next, generate a review and feedback that should be constructive, helpful, and concise, focusing on the {metric} of the method.
- Finally, provide a score on a 5-point Likert scale, with 1 being the lowest, please ensuring a discerning and critical evaluation to avoid a tendency towards uniformly high ratings (4-5) unless fully justified:

The criteria for {metric} evaluation: {criteria}
I am going to provide the proposed method with its code implementation, as follows:

Proposed method: {Method}
Code implementation:

{code}
After your evaluation of the above content, please respond **only** with a valid JSON object in the following format: { "Review": "Your review here", "Feedback": "Your feedback here", "Rating": "Your rating here" }

---

# F  Agent Trace Analysis

In this section, we analyze the agent traces for *Gemini-exp-1206* across different tasks. We collect trajectories across 7 tasks with 8 runs for each task, resulting in a total of 56 trajectories.

## F.1  Error Type Categorization

We categorize *Gemini-exp-1206*'s actions on MLRC-Bench tasks into two main types:

- **Non-execute Steps**: Steps where the action "Execute Script" was not invoked.
- **Execute Steps**: Steps where the action "Execute Script" was invoked.

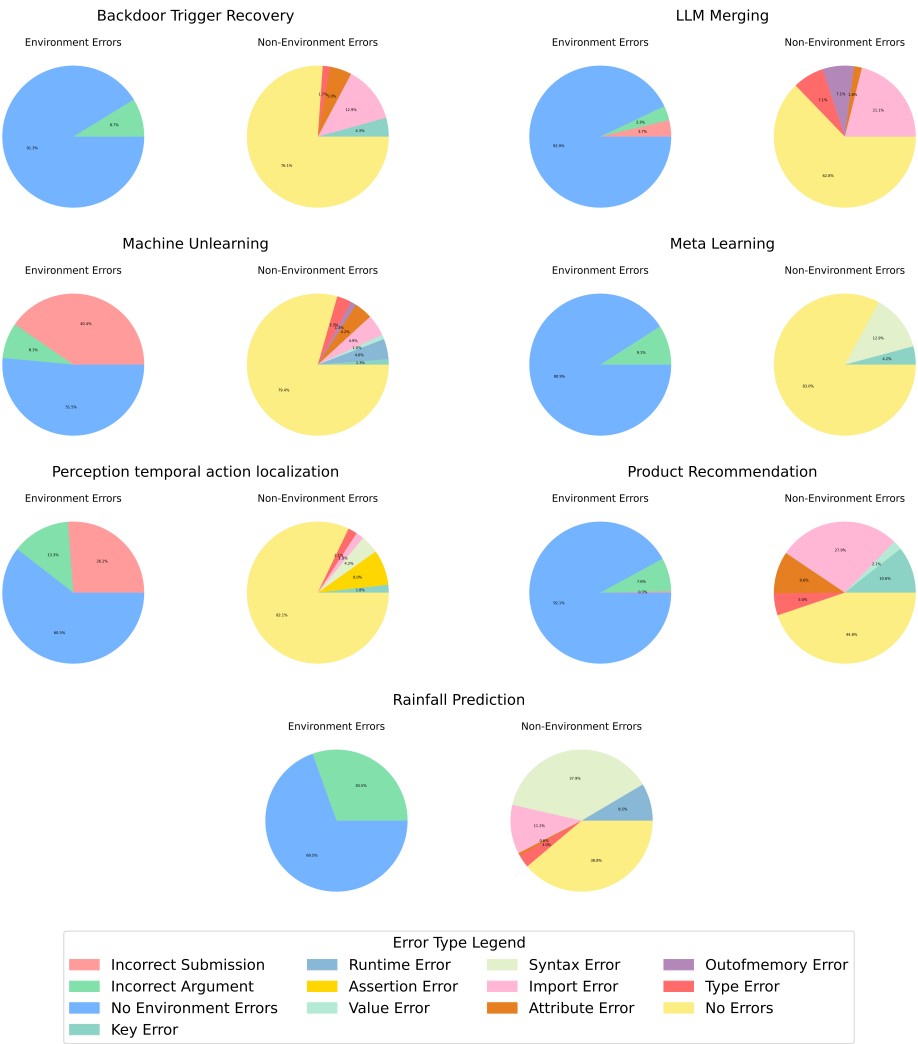

Figure 11: Distribution of environment and non-environment errors across different MLRC-Bench tasks. Each task is represented with two pie charts: one for errors related to the environment (e.g., submission issues, argument mismatches) and another for non-environment errors (e.g., runtime failures, memory issues) .

Non-execute errors are further classified into `incorrect argument` and `incorrect submissions`. Execute errors include `key errors`, `value errors`, `type errors`, `assertion errors`, `runtime errors`, `attribute errors`, `out-of-memory errors`, `import errors`, and `syntax errors`.

We briefly explain the errors encountered:

- **EnvError**: Occurs when submissions do not match the leaderboard records, files are missing, or arguments are passed incorrectly.
- **KeyError**: Results from passing incorrect argument names or not registering methods.
- **ValueError**: Triggered by invalid parameters, such as an improper learning rate or an empty parameter list.
- **TypeError**: Occurs from unexpected keyword arguments.
- **AssertionError**: Occurs when conditions such as shape compatibility or divisibility are not met.
- **RuntimeError**: Typically related to tensor shape issues.

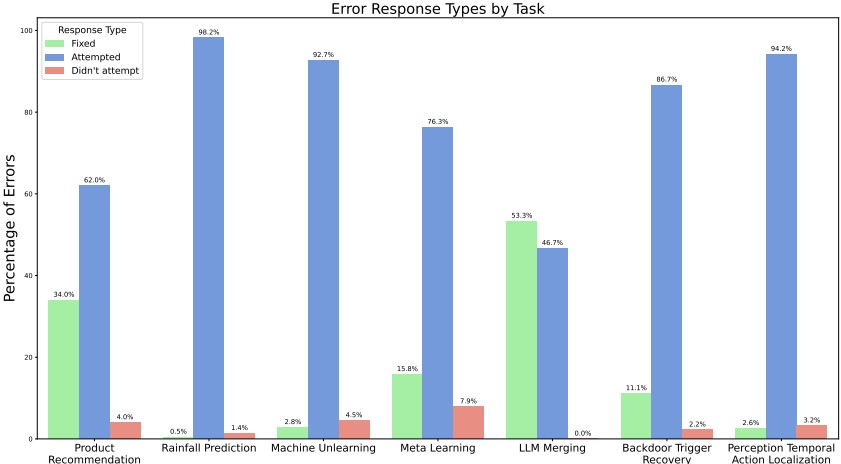

Figure 12: Error response distribution across tasks. For each task, errors are classified as *Fixed* (fully resolved), *Attempted* (partially addressed), or *Didn't attempt* (unresolved). These labels were assigned by GPT-4o-mini after evaluating each error along with all its subsequent steps (action, reflection, thought, and observation).

- **AttributeError**: Happens when a required attribute is missing.

- **OutofmemoryError**: Indicates a CUDA out-of-memory condition.

- **ImportError**: Occurs when a module cannot be imported.

- **SyntaxError**: Triggered by syntax issues, such as a missing comma.

Figure 11 shows that *Gemini-exp-1206* successfully completes a considerable number of steps without errors, yet its performance varies noticeably across tasks. In particular, while Meta Learning displays relatively few issues, Rainfall Prediction exhibits a higher frequency of "hallucination" based errors such as incorrect argument handling, non-existent file references, and invalid parameter choices. This discrepancy indicates that certain tasks present greater challenges for the model, likely due to more complex or less familiar contexts.

Within the **Execute Steps**, the most frequent error types are import, value, and type errors, reflecting a tendency to reference nonexistent modules, pass invalid parameters, or supply arguments of the wrong data type. On the **Non-execute Steps** side, incorrect arguments remain a recurring challenge, showing another case where the agent seems to be "hallucinating" the argument names.

Taken together, these findings highlight the generally robust completion of tasks, but also highlight the need to refine the agent's internal checks to reduce parameter mismatches and submission errors. Strengthening agent self-verification strategies could help mitigate hallucinations and further align its outputs with the intended specifications of each task.

## F.2 Error Response Distribution

Figure 12 presents an overview of how errors are handled across the seven tasks, highlighting the proportion of errors that were fully resolved (*Fixed*), partially addressed (*Attempted*), or left unaddressed (*Did not attempt*). These groupings were derived by passing each error along with all its next steps— containing its *action*, *reflection*, *thought* and *observation*— to GPT-4o-mini, and the error was then labeled based on whether it was successfully resolved, partially addressed, or not addressed at all.

From Figure 12, we observe notable variations in error-handling effectiveness across tasks. Specifically, *LLM Merging* demonstrates the highest proportion of fully resolved (*Fixed*) errors, indicating more effective resolution strategies, whereas *Rainfall Prediction*, *Backdoor Trigger Recovery*, *Machine Unlearning*, and *Perception Temporal Action Localization* predominantly exhibit errors that are only partially addressed (*Attempted*). Meanwhile, *Meta Learning* has the largest share of er-

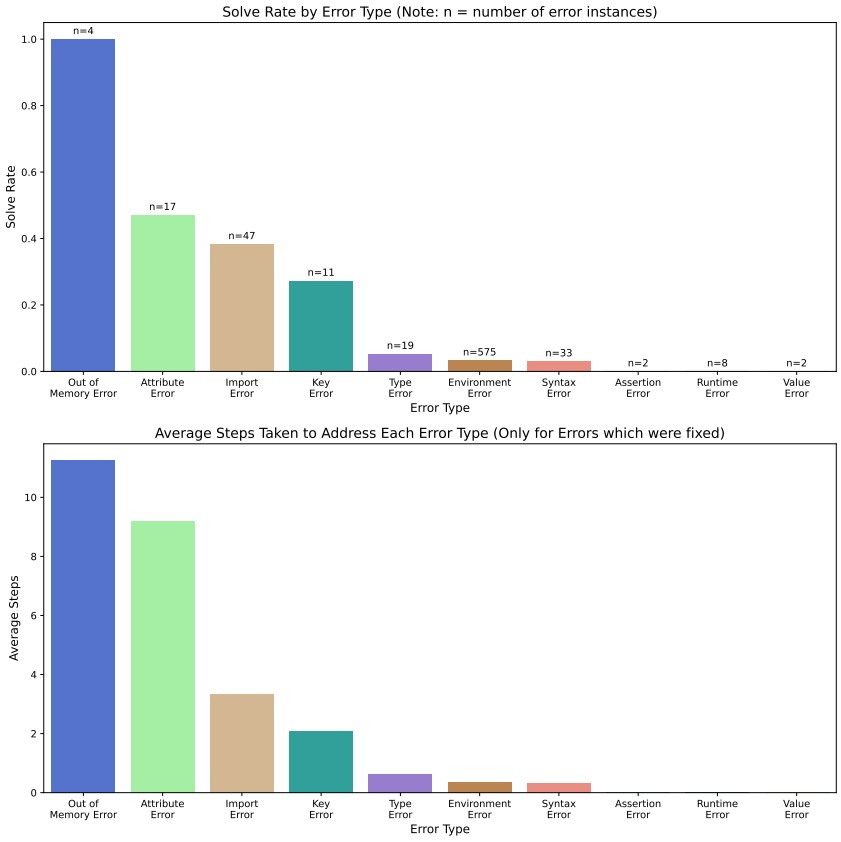

Figure 13: Solve rate and average steps taken for resolving various error types. The top chart shows the proportion of errors successfully resolved (*Solve Rate*), annotated with the total number of instances per error type. The bottom chart illustrates the average number of steps required to achieve resolution, only errors which were fixed were used to calculate average steps.

rors categorized as *Did not attempt*. These distinctions highlight task-specific differences in error management.

We also observe a consistently high percentage of errors categorized as *Attempted* across nearly all tasks, indicating that the agent often struggles to fully resolve errors. This broadly suggests challenges in the agent's comprehension or planning capabilities when addressing complex errors, potentially pointing to difficulties in fully interpreting the underlying problem or effectively formulating corrective actions. Additionally, the notable variability in fully resolved (*Fixed*) and unaddressed (*Did not attempt*) errors across tasks implies that certain tasks inherently pose greater cognitive complexity or ambiguity, further exacerbating the agent's difficulty in error resolution. The prompts used for this annotation are shown in Appendix I.

## F.3 Error Solve Rate

To further expand on this analysis of errors, we also show which error types are more effectively resolved and highlight their associated complexity during task execution. Errors which were categorised as *Fixed* are treated as solved while errors belonging to the other two categories are treated as unsolved. Using the prompt in Appendix I, we also had GPT-4o-mini return the step at which the error was fixed for those that were categorised as fixed.

Figure 13 provides insights into the solve rates across different error types, revealing variability in the agent's efficiency in resolving specific errors. Among the error types, *Out of Memory Error* achieved the highest solve rate, suggesting that these errors are relatively straightforward for the agent to diagnose and address. In contrast, *Syntax Errors* and *Environment Errors* demonstrated

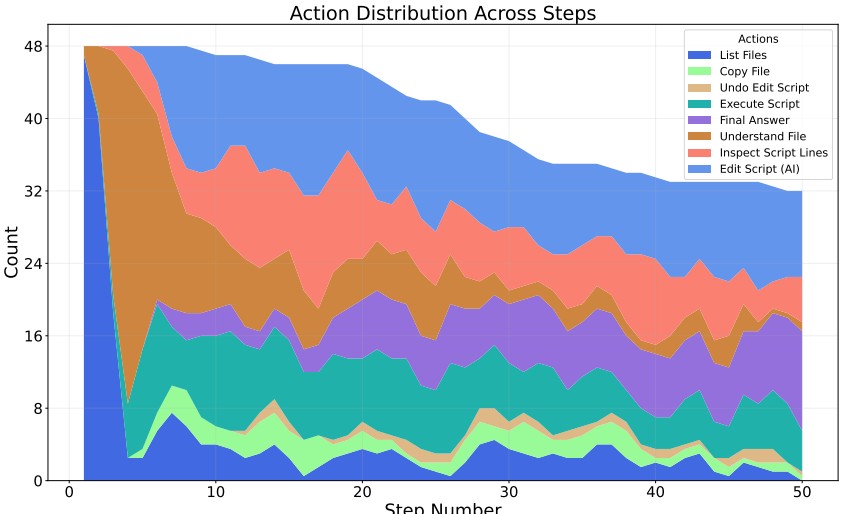

Figure 14: Distribution of Actions Taken Across Steps. This visualization depicts how frequently different types of actions were taken at each step by the agent.

lower resolution rates, while *Value Error*, *Runtime Error*, and *Assertion Error* were never fixed, highlighting their inherent complexity or ambiguity.

Additionally, the average number of steps taken to resolve errors further underscores these differences. Notably, *Out of Memory Errors* required the highest average number of steps, indicating that, although these errors are ultimately resolved at a high rate, their resolution involves a complex, multi-step process. Conversely, when *Syntax Errors* and *Environment Errors* are fixed, they tend to be resolved more quickly, suggesting that these issues, while more challenging to fix overall, can be diagnosed and corrected with fewer steps when addressed successfully.

### F.4 Per-Step Action Distribution

Figure 14 presents how frequently each action (*List Files*, *Understand File*, *Edit Script (AI)*, *Execute Script*, *Copy File*, *Undo Edit Script*, *Inspect Script Lines*, and *Final Answer*) is used over the maximum allowed 50 steps. This breakdown helps us observe when the agent transitions from environment exploration to iterative code refinement and debugging. In particular, *Rainfall Prediction* was not used for this analysis, as it was run for 100 steps.

Early steps are dominated by environment-inspection actions, particularly *List Files* and *Understand File*, which give the agent context about available files and their contents. As the trajectory progresses, the agent increasingly relies on *Edit Script (AI)* and *Execute Script* for iterative code modifications and testing, while *Inspect Script Lines* helps to target debugging. *Undo Edit Script* is used far less frequently, suggesting that the agent rarely reverts to a previous state. This pattern highlights an iterative development approach, but also indicates that the agent may underutilize rollback strategies when encountering errors. Although *Final Answer* typically signals the end, some runs exhibit early submission, indicating missed opportunities for further refinements.

### F.5 Per-Step Stage Distribution

In this section, we analyze the per-step stage distribution, categorizing the steps into seven stages based on GPT-4o annotations: `Understanding & Exploration`, `Baseline Assessment`, `Problem Analysis & Idea Generation`, `Implementation`, `Debugging & Error Handling`, `Experimental Refinement`, and `Final Evaluation & Submission`. Each step in the agent's trajectory—comprising its *Reflection*, *Thought*, *Action Input*, and *Action*—was labeled by GPT-4o, which matched the step content to the most relevant stage criteria.

Figure 15 visualizes the distribution of these seven stages over the course of the maximum allowed 50 steps. In particular, *Rainfall Prediction* was not used for this analysis, as it was ran for 100 steps.

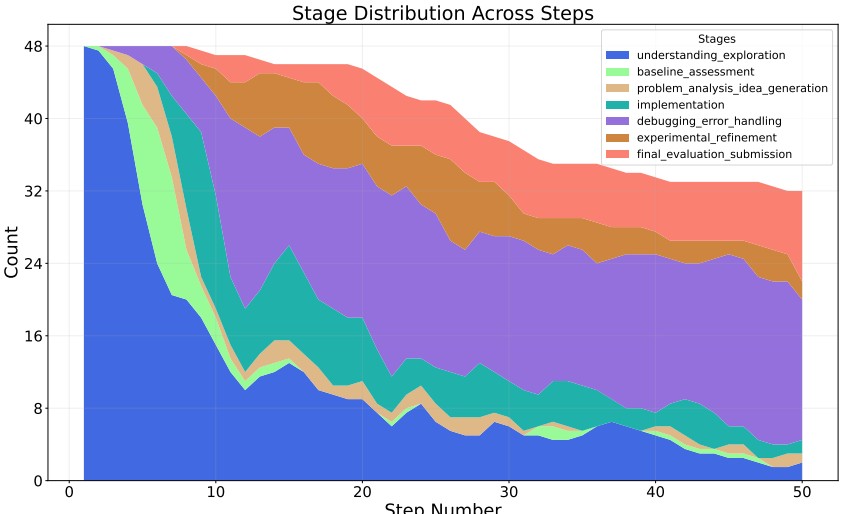

Figure 15: Stage distribution across each step, annotated using GPT-4o and grouped into seven distinct stages to illustrate shifts in task focus and activity over the course of all tasks.

The early steps are predominantly labeled **Understanding & Exploration**, reflecting the initial focus of the agent on examining files, reviewing the environment, and clarifying task requirements. A smaller portion of these early steps is allocated to **Baseline Assessment**, where the agent measures the performance of the unmodified solution to establish a reference point.

As the agent progresses, the distribution shifts noticeably toward **Implementation**, reflecting a transition from initial passive information gathering to active code modifications. Notably, the agent dedicates very few steps to **Problem Analysis & Idea Generation**, suggesting a rapid move from conceptual planning to execution. This change is often accompanied by a surge in **Debugging & Error Handling** steps, as newly introduced modifications lead to runtime or logical errors that must be diagnosed and fixed. The close interplay between **Implementation** and **Debugging & Error Handling** underscores the iterative nature of the agent's development process.

Interestingly, it should be noted that the agent continues to spend a substantial number of steps in the **Understanding & Exploration** stage. This ongoing emphasis highlights the inherent complexity and cognitive demands of repository-level tasks, which often require extensive file navigation and conceptual understanding.

Toward the latter steps, a subset of runs proceeds to **Experimental Refinement**, engaging in repeated re-runs, parameter tuning, and exploring alternative strategies to optimize performance. However, in many cases, the agent transitions relatively quickly to **Final Evaluation & Submission**. This early move towards final submission implies potential underuse of iterative enhancement cycles, indicating an area for improvement in the agent's approach. The prompts used for this annotation are shown in Appendix H.

### F.6 Stage Timelines

Using the stage annotation from the previous section, we now extend our analysis by visualizing stage timelines for each task and run. Figure 16 depict the duration the model spends in each stage, ranging from **Understanding & Exploration** to **Final Evaluation & Submission**, with block widths proportional to the time allocated. The overall run durations are also displayed, providing context for the stage-wise time distribution. Notably, *Rainfall Prediction* was not used for this analysis, as it was ran for 10 hours.

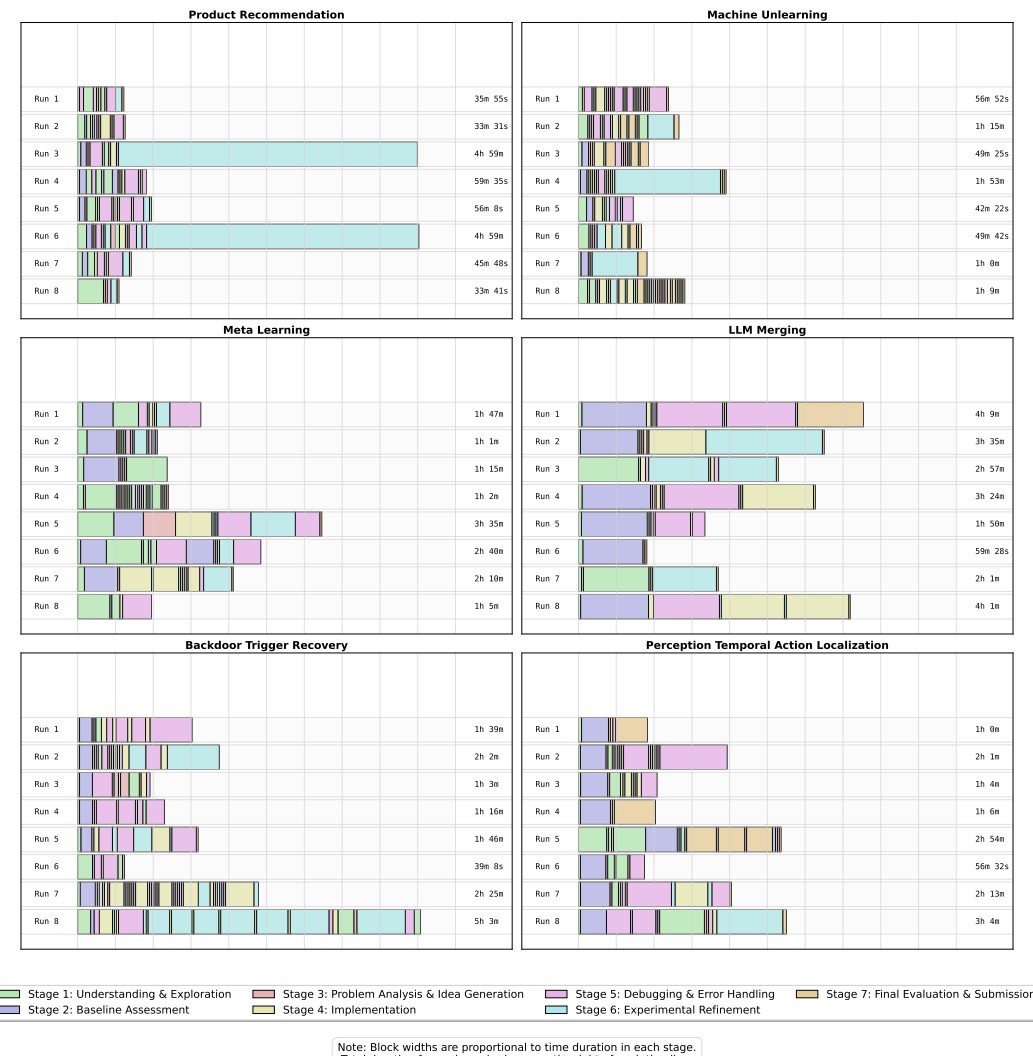

Figure 16: Combined stage timelines across multiple tasks. Each timeline represents an individual run, with block widths proportional to the time spent in each stage. The total duration of each run is shown on the right.

Table 6: Highest Capability Levels Across Experimental Runs for Evaluated Agents. This table reports the highest capability level, as defined by L1–L8 metric, achieved by each agent over eight runs across seven distinct tasks. Each run is assigned a numeric score corresponding to its level (e.g., L6 = 6, L5 = 5, and so on).

| Agent | temporal -action-loc | llm -merging | product -rec | rainfall -pred | meta -learning | machine -unlearning | backdoor -trigger |
|---|---|---|---|---|---|---|---|
| MLAB (gemini-exp-1206) | 4 | 5 | 5 | 6 | 4 | 6 | 6 |
| MLAB (llama3-1-405b-instruct) | 5 | 3 | 5 | 6 | 3 | 6 | 6 |
| MLAB (o3-mini) | 5 | 3 | 5 | 6 | 3 | 5 | 6 |
| MLAB (claude-3-5-sonnet-v2) | 5 | 5 | 5 | 6 | 3 | 4 | 6 |
| MLAB (gpt-4o) | 5 | 5 | 5 | 6 | 3 | 4 | 6 |
| Human Idea + MLAB (gpt-4o) | 5 | 3 | 5 | 6 | 3 | 6 | 6 |
| CoI-Agent (o1) Idea + MLAB (gpt-4o) | 5 | 3 | 5 | 6 | 3 | 6 | 5 |

### F.7 Capability Level

We categorize each experimental run into one of eight capability levels (L1 to L8) based on its performance relative to both a baseline and the top human solution. Definitions of each level are described below:

- **L1: No Valid Output.** The agent fails to generate any valid evaluation outputs on either the development or test set, indicating a complete inability to produce usable predictions.

- **L2: Test Submission Failure.** The agent processes the development set but fails to produce a valid submission on the test set, meaning that while some processing occurs, the pipeline does not yield a final result.

- **L3: Unimproved but Valid.** The agent produces valid predictions for both the development and test sets yet remains below the baseline performance throughout the run.

- **L4: Overfitting.** The agent outperforms the baseline on the development set but falls short on the test set, suggesting that the model may have overfitted to the development data.

- **L5: Baseline-Comparable.** The agent's test performance surpasses the baseline but remains under 5% of the margin by which the top human solution exceeds the baseline. In this range, performance is very close to the baseline level.

- **L6: Notable Gains.** The agent's test performance exceeds the baseline by an improvement margin between 5% and 50% of the gap between the baseline and the top human solution. In practical terms, this level is our "success" scenario because it indicates the agent has closed a meaningful portion of the gap above the baseline.

- **L7: Near Human-Level.** The agent captures between 50% and 100% of the improvement margin from the baseline to the top human solution, demonstrating that the performance is approaching that of the best human score.

- **L8: Superhuman.** The agent exceeds top human performance not only by delivering superior quantitative results, but also by demonstrating the creative ability to generate novel ideas and implement them effectively. This level signifies that the agent can innovate beyond established human benchmarks.

This metric places an agent's performance on a tiered scale, relative to both the baseline and the top human solution, ensuring that any level of improvement (or lack thereof) is still meaningfully captured, even when the agent falls short of surpassing the baseline. Table 6 shows that rainfall-pred and backdoor-trigger are relatively easier tasks in our benchmark as the agent can achieve a meaningful improvement over the baseline (L6), though still far behind the human. The other tasks appear to be very difficult for all agents, as they cannot achieve capability levels greater than L5.

## G    Agent Code Samples

Here we show the two examples of solutions generated by LLM agents mentioned in Section C.1.

```
1   # High-Level Overview:
2   # Purpose: This script merges multiple HuggingFace model checkpoints
        into a single base model by computing the median
3   #          of each corresponding parameter across different
        checkpoints.
4   # Methodology:
5   #    1. Load multiple HuggingFace model checkpoints along with their
        configurations.
6   #    2. For every parameter in the models, stack the corresponding
        tensors along a new dimension and compute the median value.
7   #       The median is computed in float precision and then cast back
        to the original data type.
8   #    3. Load the base model and its tokenizer.
9   #    4. Update the base model's parameters with the aggregated median
        values and set the model to evaluation mode.
10  #
11  # Key Steps:
```

```python
12  #     - Load checkpoints and configurations.
13  #     - Iterate over each parameter, compute the median across
        checkpoints with careful type conversion.
14  #     - Load the base model and tokenizer.
15  #     - Update the model state and prepare it for inference.
16
17  import torch
18  from methods.BaseMethod import BaseMethod
19  from peft import get_peft_model, set_peft_model_state_dict
20
21  class MedianMethod(BaseMethod):
22      """
23      MedianMethod performs the merging of multiple checkpoint models
            by computing the median of each parameter.
24
25      This class extends the BaseMethod to load HuggingFace
            checkpoints, compute a robust median-aggregated state,
26      and update the base model accordingly.
27      """
28
29      def __init__(self, name):
30          """
31          Initialize the MedianMethod instance.
32
33          Parameters:
34              name (str): The identifier for this method instance.
35
36          Returns:
37              None
38          """
39          # Call the parent BaseMethod's initialization method.
40          super().__init__(name)
41
42      def run(self):
43          """
44          Execute the merging pipeline and load the updated base model.
45
46          Detailed Steps:
47              1. Load HuggingFace model checkpoints and configurations.
48                  - Uses a helper function to populate
                        'self.loaded_models' with state dictionaries from
                        different checkpoints.
49              2. Merge the checkpoints by iterating over each parameter
                     key:
50                  - For each parameter, retrieve the corresponding
                        tensor from every loaded model.
51                  - Detach the tensor from the computation graph and
                        move it to CPU.
52                  - Stack these tensors along a new dimension (dim=0) to
                        form a single tensor.
53                  - Convert the stacked tensor to float for precise
                        median computation,
54                    compute the median along the new dimension, and cast
                        the result back to the original data type.
55              3. Load the base model's architecture and its tokenizer
                        via helper functions.
56              4. Update the base model's parameters with the merged
                        state dictionary and set it to evaluation mode.
57
58          Returns:
59              torch.nn.Module: The updated base model, now containing
                    the median-aggregated parameters.
60          """
61
```

```
62          # Step 1: Load HuggingFace model checkpoints and
                configurations.
63          # This helper function populates self.loaded_models with
                state dictionaries from different checkpoints.
64          super()._load_huggingface_models_and_configs()
65
66          # Step 2: Merge checkpoints by computing the median of each
                parameter across all loaded models.
67          # Retrieve all model state dictionaries as a list.
68          all_models = list(self.loaded_models.values())
69
70          # Assume all models share the same architecture; extract
                parameter names from the first model.
71          all_parameter_names = all_models[0].keys()
72
73          # Iterate over each parameter name.
74          for parameter_name in all_parameter_names:
75              # Retrieve the parameter tensor for the current parameter
                    from each model,
76              # detaching from its computation graph and moving it to
                    CPU to ensure consistency in merging.
77              param_list = [model[parameter_name].detach().cpu() for
                    model in all_models]
78
79              # Stack the tensors along a new dimension (dim=0) to
                    create a single tensor.
80              stacked_params = torch.stack(param_list, dim=0)
81
82              # Compute the median across the new dimension.
83              # The tensor is first cast to float for precision during
                    the median computation,
84              # then the median result is cast back to the original
                    data type.
85              median_value = torch.median(stacked_params.float(),
                    dim=0)[0].to(stacked_params.dtype)
86
87              # Save the computed median tensor in the merged_model
                    dictionary for later use.
88              self.merged_model[parameter_name] = median_value
89
90          # Step 3: Load the base model's architecture and its
                tokenizer.
91          # These helper functions initialize the base model and
                configure its tokenizer.
92          self._load_base_model()
93          self._load_tokenizer()
94
95          # Step 4: Update the base model with the merged parameters.
96          # Load the merged state dictionary into the base model.
97          self.base_model.load_state_dict(self.merged_model)
98
99          # Set the base model to evaluation mode to disable
                training-specific layers like dropout.
100         self.base_model.eval()
101
102         # Return the updated base model ready for inference.
103         return self.base_model
```

Listing 1: Median Merging Solution by MLAB (o3-mini) for the LLM Merging Challenge

```
1   """
2   Gradient Ascent Unlearning Algorithm
3   --------------------------------
4   Purpose: Selectively unlearn specific training samples while
        retaining knowledge of others
```

```
5   Methodology : Two - phase approach combining gradient ascent and
        fine - tuning
6   Key Steps :
7   1. Phase 1: Gradient ascent on forget set to maximize loss
        ( unlearning )
8   2. Phase 2: Fine - tuning on retain set to restore desired knowledge
9   """
10
11  from copy import deepcopy
12  import torch
13  from torch import nn , optim
14  from methods . BaseMethod import BaseMethod
15
16  DEVICE = 'cuda' if torch . cuda . is_available () else 'cpu'
17
18  class GradientAscentUnlearning ( BaseMethod ):
19      def __init__ ( self , name ):
20          """ Initialize the unlearning method
21
22          Args :
23              name : Name identifier for the method
24          """
25          super (). __init__ ( name )
26
27      def get_name ( self ):
28          """ Return the name of the unlearning method
29
30          Returns :
31              String identifier for the method
32          """
33          return " gradient_ascent_unlearning "
34
35      def run ( self , net , retain_loader , forget_loader , val_loader ):
36          """ Implement two - phase unlearning using gradient ascent and
                fine - tuning
37
38          Args :
39              net : The model to be unlearned
40              retain_loader : DataLoader for retained training data
41              forget_loader : DataLoader for data to be forgotten
42              val_loader : DataLoader for validation data
43
44          Returns :
45              The unlearned model
46          """
47          criterion = nn . CrossEntropyLoss ()
48
49          # Phase 1: Gradient Ascent on forget set
50          optimizer_forget = optim . SGD ( net . parameters () , lr =0.0001 ,
51                                  momentum =0.9 , weight_decay =5e -4)
52
53          for epoch in range (2): # 2 epochs for forgetting
54              net . train ()
55              for batch_idx , sample in enumerate ( forget_loader ):
56                  # Handle different data formats ( dict vs tuple )
57                  if isinstance ( sample , dict ):
58                      inputs = sample [" image "]
59                      targets = sample [" age_group "]
60                  else :
61                      inputs , targets = sample
62                  inputs , targets = inputs . to ( DEVICE ) ,
                        targets . to ( DEVICE )
63
64                  optimizer_forget . zero_grad ()
65                  outputs = net ( inputs )
```

```
66              loss = criterion(outputs, targets)
67              # Multiply gradients by -1 for gradient ascent
68              loss.backward()
69              for param in net.parameters():
70                  param.grad = -param.grad
71              optimizer_forget.step()
72
73          # Phase 2: Fine-tune on retain set
74          optimizer_retain = optim.SGD(net.parameters(), lr=0.01,
75                                  momentum=0.9, weight_decay=5e-4)
76          scheduler =
              torch.optim.lr_scheduler.CosineAnnealingLR(optimizer_retain,
              T_max=1)
77
78          # 5 epochs for fine-tuning
79          for epoch in range(5):
80              net.train()
81              for batch_idx, sample in enumerate(retain_loader):
82                  # Handle different data formats (dict vs tuple)
83                  if isinstance(sample, dict):
84                      inputs = sample["image"]
85                      targets = sample["age_group"]
86                  else:
87                      inputs, targets = sample
88                  inputs, targets = inputs.to(DEVICE),
                      targets.to(DEVICE)
89
90                  optimizer_retain.zero_grad()
91                  outputs = net(inputs)
92                  loss = criterion(outputs, targets)
93                  loss.backward()
94                  optimizer_retain.step()
95              scheduler.step()
96
97          net.eval()
98          return net
```

Listing 2: Gradient Ascent Unlearning Solution by MLAB (claude-3-5-sonnet-v2) for the Machine Unlearning Challenge

# H   Prompt for Stage Annotation

---

**Prompt for LLM Stage Annotator**

You are a researcher. You are given the following trace of an AI agent working on ML research challenges:

`{output_json_str}`

Your task is to analyze every step in the trace and assign a stage to each step. Use the following 7 stages. For each stage, use the reasoning guidelines provided to decide if a step belongs to that stage.

1. Understanding & Exploration:
- Description: Investigate the problem statement, explore the codebase, review data files, and understand evaluation metrics. This stage is about gathering context and building a solid grasp of the task and environment.
- Reasoning Guideline: Assign a step to this stage if it focuses on examining available resources, reading documentation or files, exploring the code structure, or otherwise building an initial understanding of the project.

2. Baseline Assessment:
- Description: Evaluate the unmodified baseline solution's performance to collect performance metrics and establish a reference benchmark.
- Reasoning Guideline: Assign a step to this stage if it focuses on measuring the performance of the original, unaltered solution, collecting data for baseline comparison, and ensuring the initial performance level is documented. Do not assign a step to this stage if it executes the solution after changes have been made.

3. Problem Analysis & Idea Generation:
- Description: Analyze the baseline results to identify shortcomings and brainstorm potential improvements or alternative strategies.
- Reasoning Guideline: Assign a step to this stage if it is centered on evaluating baseline outcomes, identifying issues, or generating ideas and strategies for potential improvements.

4. Implementation:
- Description: Develop and integrate the proposed modifications into the codebase by editing, extending, or refactoring the existing solution.
- Reasoning Guideline: Assign a step to this stage if it involves writing new code, modifying existing code, or integrating changes aimed at improving the solution.

5. Debugging & Error Handling:
- Description: Identify, isolate, and fix any errors or unexpected behaviors introduced during implementation to ensure the solution runs reliably.
- Reasoning Guideline: Assign a step to this stage if it is focused on diagnosing problems, investigating error messages, or making corrections to ensure proper functionality.

6. Experimental Refinement:
- Description: Re-run experiments on an already implemented solution and iteratively test various configurations, tune parameters, and compare alternative approaches to upgrade performance.
- Reasoning Guideline: Assign a step to this stage if it involves re-executing or adjusting an implemented solution, making upgrades and modifications to improve performance after the initial implementation has been established.

7. Final Evaluation & Submission:
- Description: Conduct a comprehensive evaluation of the refined solution against benchmarks and prepare the solution for final submission.
- Reasoning Guideline: Assign a step to this stage if it involves performing a final, thorough evaluation of the solution's performance, verifying that all improvements meet the required criteria, and preparing for submission.

---

Your response must be a JSON object where the keys are the step numbers (as strings) and the values are the corresponding stage numbers (from 1 to 7) that best describe the agent's activity at that step.

IMPORTANT: When assigning a stage, review the steps before and after each step to understand the broader context.

IMPORTANT: The original trace has `{original_step_count}` steps. Your response MUST contain exactly `{original_step_count}` keys, numbered from "1" to "`{original_step_count}`".

Example output format:

```
{
    "1": 1,
    "2": 1,
    "3": 4,
    "4": 6,
    "5": 7,
    "6": 7,
    ...
}
```

## I  Prompt for Error Response Annotation

Prompt for LLM Error Response Annotator

Below is a detailed chain-of-thought from an agent after encountering an error message:

`{error_step}`

Based on the provided debugging steps, classify the agent response regarding the error as follows:

1 -> Fixed the error: The agent identified the issue and implemented a solution that resolved the error.
2 -> Tried to fix the error but didn't: The agent attempted to address the error but the fix was not successful.
3 -> Didn't even try to fix the error and just went off doing something else: The agent did not directly attempt to resolve the error but instead focused on other tasks unrelated to fixing it.

If the error was fixed (status -> 1), also identify which step number was the error fixed at.

Return a JSON with two fields:
- Status: The number (1, 2, or 3) corresponding to the classification
- FixedAtStep: The step number where the error was fixed (only if Status is 1, otherwise null)

## J  Impact Statement

The ability of AI agents to perform high-quality ML research could accelerate breakthroughs in critical domains such as healthcare, climate modeling, and AI safety. However, agents capable of autonomously generating novel methods at scale may also amplify risks if their outputs outpace human understanding or oversight. While current agents underperform human researchers, MLRC-BENCH highlights the need for ongoing monitoring of their capabilities to ensure alignment with ethical standards and societal goals. By releasing this benchmark, we aim to enhance transparency and encourage the development of safer, more reliable AI research agents. We caution against deploying such systems without robust safeguards and urge the community to prioritize evaluations that balance innovation with accountability.

