# OpenReview forum: "MLRC-Bench: Can Language Agents Solve Machine Learning Research Challenges?"
_NeurIPS.cc/2025/Datasets_and_Benchmarks_Track — NeurIPS 2025 Datasets and Benchmarks Track poster_

### Official Review · Reviewer_3orb · 2025-07-02

**Rating:** 4
**Confidence:** 3

**Summary:**

The authors curated seven competition tasks from recent machine learning conferences to construct MLRC-Bench. They evaluated five models as potential backbones for the MLAB agent, reporting their performance relative to human solutions. Additionally, the authors investigated whether human ideas or chain-of-thought (CoT) reasoning could enhance model performance. They also find that LLM-as-Judge fails to reliably assess the quality of these ideas.

**Dataset Code Accessibility:**

Yes

**Ethical Considerations:**

No, there are no or only very minor ethics concerns

**Final Justification:**

Overall, this work is a solid contribution to the LLM Agents' community. My concerns on limitations 2&3 have been solved.
However, I still maintain my opinion on the limited benchmark scope. Though the authors argue that recent works have a similar size, I also should point out that the OpenAI MLE-Bench has a much larger size, which greatly harms the contribution of this work.
Considering the rebuttal process, I raise my score to 4.

**Limitations Weaknesses:**

1. While similar works like MLE-Bench contain 65 competitions, this work only includes 7 competitions, which makes me wonder about the reliability of the results.
2. The metric of relative improvement to the human solution is wired and not intuitive to readers. It's really hard to see the actual performance of different methods.
3. The inference-time scaling evaluation is conducted on a relatively small scale (8 trials at most), which makes the result not very convincing. Besides, the *Human Idea + MLAB* settings look like a straight line, can the author explain it more detailly?

**Strengths Contributions:**

1. The author provides the essential codes to replicate the results. The collected competition is high-quality.
2. The author involves human ideas into the benchmark and provides inference-time scaling under different settings.

---

> ### Author Rebuttal · Authors · 2025-07-31
>
> We thank the reviewer for their thoughtful and constructive feedback; below we address each of the points you raised.
>
> ***Re: While similar works like MLE-Bench contain 65 competitions, this work only includes 7 competitions, which makes me wonder about the reliability of the results.***
>
> Our MLRC-Bench features 7 research-driven tasks in total, a number comparable to several recent benchmarks for AI research agents. For example, MLAgentBench [1] has 13 tasks, MLGym-Bench [2] has 13 tasks, and RE-Bench [3] also contains 7. However, MLRC-Bench spans a broader range of research domains and addresses more cutting-edge problems (Lines 66–68), with an additional emphasis on **repository-level code complexity**. Moreover, MLRC-Bench is designed to *evolve dynamically* as new ML competitions emerge, with a structure that explicitly supports the addition of future tasks to ensure continued alignment with frontier challenges in ML research (Line 108).
>
> The current task set is not only comparable in size but also carefully curated to reflect **high-impact, unsolved** research challenges drawn from recent ML conference competitions. Each task requires **genuine algorithmic creativity**, rather than superficial improvements such as model tuning or ensembling. Unlike existing benchmarks such as MLE-Bench, which prioritize breadth and generalization across many tasks, MLRC-Bench emphasizes depth—*solving even a single task represents meaningful scientific progress*. In this regime, evaluation is not merely about averaging performance across multiple tasks but about assessing whether a system can successfully tackle a **complex, real-world research problem**.
>
>
>
> [1] Qian Huang, Jian Vora, Percy Liang, and Jure Leskovec. MLAgentBench: Evaluating Language Agents on Machine Learning Experimentation. In Forty-first International Conference on Machine Learning, ICML 2024, Vienna, Austria, July 21-27, 2024.
>
> [2] Deepak Nathani, Lovish Madaan, Nicholas Roberts, Nikolay Bashlykov, Ajay Menon, Vincent Moens, Amar Budhiraja, Despoina Magka, Vladislav Vorotilov, Gaurav Chaurasia, Dieuwke Hupkes, Ricardo Silveira Cabral, Tatiana Shavrina, Jakob N. Foerster, Yoram Bachrach,William Yang Wang, and Roberta Raileanu. MLGym: A New Framework and Benchmark for Advancing AI Research Agents. CoRR, abs/2502.14499, 2025.
>
> [3] Hjalmar Wijk, Tao Lin, Joel Becker, Sami Jawhar, Neev Parikh, Thomas Broadley, Lawrence Chan, Michael Chen, Josh Clymer, Jai Dhyani, Elena Ericheva, Katharyn Garcia, Brian Goodrich, Nikola Jurkovic, Megan Kinniment, Aron Lajko, Seraphina Nix, Lucas Sato, William Saunders, Maksym Taran, Ben West, and Elizabeth Barnes. RE-Bench: Evaluating Frontier AI R&D Capabilities of Language Model Agents against Human Experts. CoRR, abs/2411.15114, 2024.
>
> ***Re: The metric of relative improvement to the human solution is wired and not intuitive to readers. It's really hard to see the actual performance of different methods.***
>
> Thank you for this feedback. Since tasks in MLRC-Bench vary widely in evaluation metrics (e.g., accuracy, mAP, recall) and scales, directly comparing raw scores across tasks can be misleading (Lines 160–162). We thus introduced the metric of relative improvement to the human solution to normalize performance across tasks and show a clear trend for how far agents' solutions are lagging behind the top human solutions.
>
> We agree that absolute scores are also important for reference. Table 4 in Appendix D provides the absoulute score improvements for full transparency. From that table, we see that top human competitors achieve an average 280.7% improvement over the baseline, whereas the leading language agent (Claude‑3.5‑Sonnet + MLAB) delivers only a 31.9% gain, highlighting the current gap in research‑innovation capability between humans and LLM agents.
>
> ***Re: The inference-time scaling evaluation is conducted on a relatively small scale (8 trials at most), which makes the result not very convincing.***
>
> Thank you for raising the concern about limited trial counts. As shown in Figure 8, **most tasks record zero successes across all eight trials**, so we expect that increasing trials would not change their Pass\@k curves .
>
> To probe further, we extended two representative tasks, **backdoor-trigger-recovery** and **machine-unlearning**, to **16 trials** for all three settings: MLAB-only, CoI-Agent Idea + MLAB, and Human Idea + MLAB.
>
>
> | Task                      | System                | pass@1 | pass@4 | pass@8 | pass@12 | pass@16 |
> | ------------------------- | --------------------- | ------: | ------: | ------: | -------: | -------: |
> | backdoor-trigger-recovery | MLAB                  |    0.12 |    0.45 |    0.77 |     0.95 |     1.00 |
> | backdoor-trigger-recovery | CoI-Agent Idea + MLAB |    0.19 |    0.61 |    0.90 |     0.99 |     1.00 |
> | backdoor-trigger-recovery | Human Idea + MLAB     |    **0.31** |    **0.82** |    **1.00** |     **1.00** |     1.00 |
> | machine-unlearning        | MLAB                  |    0.00 |    0.00 |    0.00 |     0.00 |     0.00 |
> | machine-unlearning        | CoI-Agent Idea + MLAB |    0.06 |    0.25 |    0.50 |     0.75 |     1.00 |
> | machine-unlearning        | Human Idea + MLAB     |    **0.12** |    **0.45** |    **0.77** |     **0.95** |     1.00 |
>
>
>
> These results underscore the importance of ideation: on both tasks, providing ideas—either from AI or human—consistently improves Pass@k compared to the MLAB-only (implementation-only) setting, with human ideas enabling faster solution discovery than AI-generated ones. This reinforces our earlier conclusion (Lines 300–302) that without true innovation capabilities, simply increasing the number of trials is unlikely to yield better performance.
>
>
> ***Re: Besides, the Human Idea + MLAB settings look like a straight line, can the author explain it more detailly?***
>
> Recall that the Pass\@k formula for $N$ attempts with $n$ successes is:
>
> $$
> \text{Pass@}k = 1 - \frac{\binom{N - n}{k}}{\binom{N}{k}}.
> $$
>
> In the Human Idea + MLAB setting, there is only 1 success across 8 trials. So when $n = 1$, this simplifies to:
>
> $$
> \text{Pass@}k = \frac{k}{N},
> $$
>
> which explains the “straight-line” appearance rather than indicating a flaw in our evaluation setup.

---

> > ### Comment · Reviewer_3orb · 2025-08-05
> >
> > The authors have resolved most of my concerns, except for the limited dataset size. I will raise my score accordingly.

---

### Official Review · Reviewer_P5Tc · 2025-07-02

**Rating:** 5
**Confidence:** 4

**Summary:**

This paper presents MLRC-Bench (Machine Learning Research Competitions)
evaluating LLM agents’ capability of solving open research problems. MLRC
incorporates real-world ML competitions drawn from ML conferences. These
competitions have public leaderboards with clear objectives for judgment, and
previous human solutions can be used as reliable baselines. MLRC provides
carefully curated task description, starter code, and human insights. Agents
are evaluated though diverse dimensions: effectiveness, efficiency, and
simplicity. MLRC is empirically shown in to challenging enough even for
today’s best-performing LLMs, with gemini-exp-1206 (Gemini 2.0 Pro) merely
outperforming humen participants by 9.6%. More analysis on scalability and
subjective evaluations are also presented and discussed.

**Additional Feedback:**

MLRC is an extensible benchmark. However, I suspect human involvement is
still necessary according to the description in this paper. How much effort is
commonly required to transform a real-world ML competition into MLRC
format?
Some outliers in Table 2 are not discussed. Why claude-3-5-sonnet-v2
significantly underperforms other baselines? While this model ranked first on
four subtasks, it becomes the worst model considering -94.7% performance
gap on the machine-unlearning subtask. Is it a reasonable outcome or caused
by some unexpected bias in implementation?

**Dataset Code Accessibility:**

Yes

**Dataset Code Comments:**

The authors have published MLRC with detailed instructions on how to set up
this benchmark

**Ethical Considerations:**

No, there are no or only very minor ethics concerns

**Limitations Weaknesses:**

It may help if MLAB, as an important framework used for evaluation, is
introduced in related works, e.g., is MLAB equipped with web searching tools?

It will be easier for readers to understand the evaluation results and gain
insights from it.

Most recent AI-driven open research frameworks are not evaluated. While it is
acceptable due to optential gap in implementation and setups, the evaluation
would be more comprehensive with these recent baselines considered.

**Strengths Contributions:**

MLRC is a challenging benchmark reflecting LLM agent’s skills in real-world
ML competitions.

MLRC can evolve as future ML competitions are introduced, keeping pace with
most recent advantages in ML communities.

MLRC effectively utilized human proposed ML competitions, which are ignored
in most previous benchmarks.

---

> ### Author Rebuttal · Authors · 2025-07-31
>
> Thank you for your thoughtful and constructive feedback!
>
> ***Re: It may help if MLAB, as an important framework used for evaluation, is introduced in related works, e.g., is MLAB equipped with web searching tools?***
>
> We appreciate the suggestion. We have briefly introduced MLAB in the Line 191-194 of our draft. To clarify, MLAB does not include any built‑in web search capability. It operates entirely through a ReAct-style loop over local resources, such as code files, the Python runtime, and its internal memory. In contrast, the other agent framework we use, **CoI-Agent**, which generates research ideas as input to MLAB—is equipped with web-based tools, including access to the Semantic Scholar API for retrieving relevant literature (Line 195-198). By evaluating CoI-Agent ideas with MLAB, we effectively examine how agents can leverage web-based retrieval to address machine learning research challenges. We will also revise the Related Work section to clarify the capabilities of MLAB and highlight the differences among various AI research agents.
>
>
> ***Re: How much effort is commonly required to transform a real-world ML competition into MLRC format?***
>
> Transforming a real-world ML competition into the MLRC format involves three main stages: (1) identifying a suitable competition from recent ML conferences, (2) running the official starter kit to verify a valid baseline score, and (3) refactoring the starter kit into the MLRC task format. Based on our experience, this process typically takes a graduate student in computer science between  **20 to 40 hours**, with the primary difficulty stemming from  **environment setup**  rather than code refactoring.
>
> To scale future contributions, we have released  [templates and detailed instructions](https://tinyurl.com/MLRC-Task-Template)  to support the community (see Line 112), and we hope to encourage broader participation in expanding the benchmark.
>
> ***Re: Some outliers in Table 2 are not discussed. Why claude-3-5-sonnet-v2 significantly underperforms other baselines? Is it a reasonable outcome or caused by some unexpected bias in implementation?***
>
> This outcome is reasonable and already discussed in Lines 213–216. While Claude 3.5 Sonnet V2 achieves strong performance on most tasks, it performs notably worse on the  **machine unlearning**  challenge. As detailed in Appendix C.1, this underperformance stems from the agent’s strategy of handling  **data removal**  and  **knowledge retention**  as separate phases rather than optimizing both objectives jointly.
>
> Specifically, the agent proposed a two-stage "Gradient Ascent Unlearning Algorithm" that first applies gradient ascent on the forget set to erase unwanted data, followed by fine-tuning on the retain set to restore useful knowledge. Although this approach is conceptually sound, it scored significantly below the baseline. This sequential treatment of competing objectives fails to maintain the necessary tradeoff between forgetting and retaining. In contrast, joint optimization, which updates both objectives simultaneously, may be more effective for balancing these goals. Therefore, the result reflects a  **genuine shortcoming in the proposed method**, rather than any bias or flaw in implementation.

---

### Official Review · Reviewer_apse · 2025-07-03

**Rating:** 4
**Confidence:** 4

**Summary:**

This paper introduces MLRC-BENCH as a benchmark to evaluate the ability of LLM-based research agents to propose and implement novel methods. Drawing on tasks from recent ML conference competitions, MLRC-BENCH enables the evaluation of both novelty and effectiveness.

**Additional Feedback:**

(1) Are there methods for examining data contamination mitigation and retrieving answers in existing 7 tasks?


(2) Why were these five metrics—validity, clarity, rigorousness, generalizability, and innovativeness—chosen for subjective evaluation? Can the inconsistency between subjective and objective metrics rigorously explain the conclusions at line 269?

**Dataset Code Accessibility:**

Yes

**Ethical Considerations:**

No, there are no or only very minor ethics concerns

**Final Justification:**

- Introducing MLRC-BENCH, a dynamic benchmark suite curated from ML conference competitions to evaluate both novelty and effectiveness of LLM agents.

- Conducting large-scale, objective and subjective evaluations for a wide array of frontier LLMs with representative agent scaffoldings.
Revealing novel conclusions such as the limitations of existing agents in generating and utilizing innovative ideas under complex tasks. The experimental analysis is highly detailed.

**Limitations Weaknesses:**

- Contribution: This benchmark comprises only 7 tasks, indicating a relatively limited scope of coverage in the tasks.


- Risk: Apart from regular updates on tasks, there is a lack of methods for addressing data contamination mitigation and retrieval of answers in existing tasks.

**Strengths Contributions:**

- Introducing MLRC-BENCH, a dynamic benchmark suite curated from ML conference competitions to evaluate both novelty and effectiveness of LLM agents.

- Conducting large-scale, objective and subjective evaluations for a wide array of frontier LLMs with representative agent scaffoldings.
Revealing novel conclusions such as the limitations of existing agents in generating and utilizing innovative ideas under complex tasks. The experimental analysis is highly detailed.

---

> ### Author Rebuttal · Authors · 2025-07-31
>
> Thank you for your thoughtful and constructive feedback!
>
> ***Re: This benchmark comprises only 7 tasks, indicating a relatively limited scope of coverage in the tasks.***
>
> Our MLRC-Bench features 7 research-driven tasks in total, a number comparable to several recent benchmarks for AI research agents. For example, MLAgentBench [1] has 13 tasks, MLGym-Bench [2] has 13 tasks, and RE-Bench [3] also contains 7. However, MLRC-Bench spans a broader range of research domains and addresses more cutting-edge problems (Lines 66–68), with an additional emphasis on **repository-level code complexity**. Moreover, MLRC-Bench is designed to *evolve dynamically* as new ML competitions emerge, with a structure that explicitly supports the addition of future tasks to ensure continued alignment with frontier challenges in ML research (Line 108).
>
> The current task set is not only comparable in size but also carefully curated to reflect **high-impact, unsolved** research challenges drawn from recent ML conference competitions. Each task requires **genuine algorithmic creativity**, rather than superficial improvements such as model tuning or ensembling. Unlike existing benchmarks such as MLE-Bench, which prioritize breadth and generalization across many tasks, MLRC-Bench emphasizes depth—*solving even a single task represents meaningful scientific progress*. In this regime, evaluation is not merely about averaging performance across multiple tasks but about assessing whether a system can successfully tackle a **complex, real-world research problem**.
>
>
> [1] Qian Huang, Jian Vora, Percy Liang, and Jure Leskovec. MLAgentBench: Evaluating Language Agents on Machine Learning Experimentation. In Forty-first International Conference on Machine Learning, ICML 2024, Vienna, Austria, July 21-27, 2024.
>
> [2] Deepak Nathani, Lovish Madaan, Nicholas Roberts, Nikolay Bashlykov, Ajay Menon, Vincent Moens, Amar Budhiraja, Despoina Magka, Vladislav Vorotilov, Gaurav Chaurasia, Dieuwke Hupkes, Ricardo Silveira Cabral, Tatiana Shavrina, Jakob N. Foerster, Yoram Bachrach,William Yang Wang, and Roberta Raileanu. MLGym: A New Framework and Benchmark for Advancing AI Research Agents. CoRR, abs/2502.14499, 2025.
>
> [3] Hjalmar Wijk, Tao Lin, Joel Becker, Sami Jawhar, Neev Parikh, Thomas Broadley, Lawrence Chan, Michael Chen, Josh Clymer, Jai Dhyani, Elena Ericheva, Katharyn Garcia, Brian Goodrich, Nikola Jurkovic, Megan Kinniment, Aron Lajko, Seraphina Nix, Lucas Sato, William Saunders, Maksym Taran, Ben West, and Elizabeth Barnes. RE-Bench: Evaluating Frontier AI R&D Capabilities of Language Model Agents against Human Experts. CoRR, abs/2411.15114, 2024.
>
>
> ***Re: Apart from regular updates on tasks, there is a lack of methods for addressing data contamination mitigation and retrieval of answers in existing tasks.***
>
> As discussed in Section 3.1 (Line 115), we mitigate contamination risks by selecting competitions whose top participants’ code solutions are  **not publicly released**, reducing the chance that implementation code appears in LLM pretraining corpora. Many tasks also represent  **open‑ended scientific problems**  whose gold solutions remain  **unknown**, making retrieval‑based shortcuts infeasible. Together with regular benchmark updates (Line 108), these design choices ensure MLRC‑Bench emphasizes  **authentic research capability**  rather than data leakage or memorization.
>
> Moving forward, we will implement a  **concept‑level novelty scoring**  module that automatically parses each solution’s core ideas into a normalized concept set, then compares this set against a growing graph of concepts extracted from published papers and competition write‑ups. By measuring the overlap between the solution’s concept subgraph and its nearest neighbors in the literature graph, we compute a “novelty score” that flags any submission whose ideas closely mirror existing work. This single metric provides an automated, principled check on whether high‑scoring agents are truly proposing new methods rather than recycling known approaches.
>
> ***Re: Why were these five metrics—validity, clarity, rigorousness, generalizability, and innovativeness—chosen for subjective evaluation? Can the inconsistency between subjective and objective metrics rigorously explain the conclusions at line 269?***
>
> We adopt those five dimensions straight from a  **peer‑reviewed rubric** [4]  that was created to assess research methods rather than the problem setup or experimental design (Line 247). That focus matches our core goal (Line 34) of evaluating an agent’s ability to propose and implement novel methods instead of simply replaying known solutions.
>
> For our analysis (not as part of the benchmark design), we use  **OpenAI’s o1**  model as a strong, independent judge as o1 was not used as the backbone model for the MLAB implementation agent. Those subjective scores exist only to probe potential biases in LLM-based judgments. The fact that o1’s subjective ratings misalign with our objective measures (the gap we highlight at Line 269) actually  **supports**  our warning that relying only on subjective judgments can yield overly optimistic conclusions.
>
> [4] Jinheon Baek, Sujay Kumar Jauhar, Silviu Cucerzan, and Sung Ju Hwang. ResearchAgent:
> Iterative Research Idea Generation over Scientific Literature with Large Language Models. CoRR,
> abs/2404.07738, 2024.

---

### Official Review · Reviewer_SMQw · 2025-07-23

**Rating:** 5
**Confidence:** 3

**Summary:**

The authors introduce MLRC-BENCH to quantitatively assess language agents’ ability to propose and implement new methods in machine learning research competitions. The first release covers 7 tasks drawn from recent top-tier venues/competitions and uses objective metrics—effectiveness, efficiency, and simplicity—to score agents. The best setup (gemini-exp-1206 + MLAB) closes only 9.3% of the gap to human top solutions, and the study shows that LLM-judge “innovation” scores correlate weakly with actual performance.

**Dataset Code Accessibility:**

Yes

**Ethical Considerations:**

No, there are no or only very minor ethics concerns

**Final Justification:**

The authors have addressed my concerns in their reponse. Therefore, I will maintain my rating.

**Limitations Weaknesses:**

Is it possible to include more LLM4AutoML work for evaluation—for example, recent systems like SELA or AutoML Agent?

**Strengths Contributions:**

1 Clear, realistic positioning: Conference competitions offer a testbed that is both novel and objectively benchmarked—avoiding Kaggle-style engineering chores with little innovation, and differing from end-to-end “write a paper” pipelines that lack solid baselines.

2 Diverse and extensible tasks: Seven tasks span LLM security, multimodal perception, few-shot learning, etc., with a stated plan to keep adding new frontier challenges.

3 Objective evaluation and anti-cheating protocol: Performance is measured by relative improvement over human baselines, with runtime/GPU memory and LOC constraints. Code snapshots and separated dev/test splits help prevent overfitting.

4 Exposes flaws in subjective judging: Quantitatively demonstrates the mismatch between LLM-judge innovation scores and true effectiveness, an important negative finding.

---

> ### Author Rebuttal · Authors · 2025-07-31
>
> Thank you for your thoughtful and constructive feedback!
>
> ***Re: Is it possible to include more LLM4AutoML work for evaluation—for example, recent systems like SELA or AutoML Agent?***
>
> AutoML frameworks such as  **SELA**  and  **AutoML-Agent**  are designed to  _automate modular, Kaggle-style ML pipelines_, typically producing a single Python script that ensembles multiple off-the-shelf models. In contrast,  **MLRC-Bench** targets research-grade challenges that demand  **repository-level code comprehension and generation**  (Line 130): agents must navigate full codebases, modify multiple files, and integrate novel algorithmic ideas.
>
> While SELA and AutoML-Agent represent important progress in standard AutoML, they are not equipped to (1) operate over multi-file research repositories or (2) propose and implement  **novel research methods beyond routine preprocessing or model-selection pipelines**. Adapting them to MLRC-Bench would require substantial reengineering of both their assumptions and execution pipelines.
>
> That said, we believe extending SELA’s tree-search-based prompting and workflow to support MLRC-style tasks presents an exciting direction for future work, potentially enabling more powerful solutions for open-ended ML research problems.

---

### Decision · Program_Chairs · 2025-09-18

**Decision:**

Accept (poster)

**Comment:**

(a) Summary of Scientific Claims and Findings
This paper introduces MLRC-Bench, a benchmark designed to evaluate the ability of LLM-based agents to propose and implement novel methods in machine learning research competitions. The benchmark includes seven tasks curated from recent top-tier ML conference competitions, spanning domains such as LLM security, multimodal perception, and few-shot learning. MLRC-Bench provides a rigorous evaluation framework with metrics of effectiveness, efficiency, and simplicity, as well as safeguards against data leakage and overfitting. Results show that current state-of-the-art LLM agents close only a small fraction of the gap to human expert performance. Importantly, the study also demonstrates that subjective LLM-as-judge innovation scores correlate poorly with objective evaluation results, cautioning against overreliance on such scoring.

(b) Strengths
The paper makes a timely and impactful contribution by introducing a benchmark that directly addresses the challenge of evaluating research-level innovation capabilities of language agents. Strengths highlighted across reviews include:

+ Novel and realistic task design, leveraging competition-style problems that demand repository-level code modification rather than single-script pipelines.
+ Objective evaluation metrics, emphasizing reproducibility, efficiency, and anti-cheating protocols.
+ Extensibility, with a dynamic structure to accommodate new tasks as competitions emerge.
+ Empirical insights, particularly the finding that subjective LLM-judge evaluations can diverge substantially from objective measures, which is an important cautionary result.
+ These align well with the DB track CFP, which seeks rigorous and responsible benchmarks that address emerging challenges in AI.

(c) Weaknesses
Reviewers raised several limitations:

- The benchmark currently covers only seven tasks, smaller in scale compared to some existing benchmarks (e.g., MLE-Bench).
- Limited discussion of data contamination checks beyond task selection, though mechanisms are in place to mitigate this risk.
- Relative improvement to human solutions was initially viewed as unintuitive by some reviewers.
- The inference-time scaling experiments were small-scale, raising questions about robustness.

These are valid but, as reviewers noted, do not outweigh the benchmark’s originality, depth, and design quality. Moreover, the authors clarified that MLRC-Bench prioritizes depth and research-grade complexity, which complements other broader but shallower benchmarks.

(d) Reasons for Acceptance
The paper fills an urgent gap in evaluating whether language agents can contribute meaningfully to machine learning research. The benchmark is carefully designed to capture dimensions of innovation and implementation, offering a rigorous baseline for future work. While the number of tasks is modest, the curated selection reflects cutting-edge challenges and is explicitly designed to grow over time. The demonstrated divergence between subjective LLM-judge ratings and objective outcomes is itself a valuable scientific finding. Taken together, the contributions are both technically solid and highly impactful, making this a strong candidate for acceptance in the DB track.

(e) Discussion and Rebuttal
The rebuttal and author–reviewer discussions were constructive:

Reviewer SMQw asked about including AutoML frameworks such as SELA or AutoML-Agent. The authors clarified these systems target Kaggle-style pipelines and are ill-suited for MLRC-Bench’s repository-level challenges, while also outlining future integration directions. The reviewer accepted this clarification and maintained a strong positive rating.

Reviewer apse raised concerns about limited task coverage, contamination risks, and subjective metric choices. The authors compared MLRC-Bench to related benchmarks, explained their contamination mitigation strategy, and justified their use of peer-reviewed subjective evaluation rubrics. These clarifications were well-received, and the reviewer maintained a positive borderline rating.

Reviewer P5Tc requested clarification on MLAB’s role, the effort required to adapt tasks, and outliers in results. The authors provided detailed explanations, including additional discussion of Claude-3.5 Sonnet’s underperformance in unlearning tasks. The reviewer’s concerns were satisfactorily resolved, and they endorsed acceptance.

Reviewer 3orb questioned the small task set, relative improvement metrics, and limited inference-time scaling. The authors provided comparative context with related benchmarks, justified their normalization metric, and extended scaling experiments to 16 trials, showing consistent conclusions. The reviewer increased their score while maintaining reservations about scope.

Overall, the rebuttal substantially strengthened the submission by clarifying design choices, adding new analyses, and situating MLRC-Bench within the broader benchmark landscape. The consensus from most reviewers supports acceptance, with the remaining concerns primarily about future extensions rather than core flaws.

Final Recommendation: I recommend acceptance. This benchmark provides a rigorous, extensible, and well-motivated platform for evaluating research innovation capabilities of LLM agents and makes an important contribution to the DB track.